# PROVABLY SAFEGUARDING A CLASSIFIER FROM OOD AND ADVERSARIAL SAMPLES

**Nicolas Atienza**
Thales cortAIx Labs
Industrial AI Laboratory SINCLAIR
LISN CNRS-INRIA
Paris-Saclay University
nicolas.atienza@thalesgroup.com

**Christophe Labreuche**
Thales cortAIx Labs
Industrial AI Laboratory SINCLAIR
Palaiseau, France
christophe.labreuche@thalesgroup.com

**Johanne Cohen**
LISN CNRS-INRIA
Paris-Saclay University
Saclay, France
jcohen@lisn.fr

**Michèle Sebag**
LISN CNRS-INRIA
Paris-Saclay University
Saclay, France
sebag@lisn.fr

## ABSTRACT

This paper aims to transform a trained classifier into an abstaining classifier, such that the latter is provably protected from out-of-distribution and adversarial samples. The proposed *Sample-efficient Probabilistic Detection using Extreme Value Theory* (SPADE) approach relies on a Generalized Extreme Value (GEV) model of the training distribution in the latent space of the classifier. Under mild assumptions, this GEV model allows for formally characterizing out-of-distribution and adversarial samples and rejecting them. Empirical validation of the approach is conducted on various neural architectures (ResNet, VGG, and Vision Transformer) and considers medium and large-sized datasets (CIFAR-10, CIFAR-100, and ImageNet). The results show the stability and frugality of the GEV model and demonstrate SPADE's efficiency compared to the state-of-the-art methods.[1]

## 1 INTRODUCTION

A key challenge in deploying learned models in real-world settings is the management of out-of-distribution (OOD) and adversarial samples. When a learned model encounters data that deviates from the training distribution or is maliciously crafted to deceive the model (Szegedy et al., 2014; Goodfellow et al., 2015), it can lead to high consequence failures , especially in high-stakes applications such as medical diagnosis, autonomous driving or risk analysis (Salehi et al., 2022; Yang et al., 2022).

The ultimate goal of machine learning is to achieve generalization, where the learned model encapsulates core concepts accurately enough to effectively handle atypical but real-world samples (Ye et al., 2021). A step towards generalization with respect to OOD samples is OOD detection, which equips the learned model with the ability to recognize atypical samples and refrain from making risky predictions. OOD detection is approached from several directions, including methods based on classification (Hendrycks & Gimpel, 2017; Liang et al., 2018; Hsu et al., 2020), reconstruction metrics (Jiang et al., 2023; Li et al., 2023), density estimation (Ren et al., 2019; Liu et al., 2021; Du et al., 2022b) and distance-based estimation (Papernot & McDaniel, 2018; Sun et al., 2022; Dziedzic et al., 2022; Ming et al., 2023) (more in Section 6). OOD detection is complicated by the fact that, to the best of our knowledge, there is no universally accepted definition of an OOD sample. The distinction between in-distribution and OOD data is inherently ambiguous, as different domain experts may classify the same sample differently based on their understanding and experience (Idrissi et al., 2023). Therefore, the validation of OOD detection methods relies heavily on experimental studies using well-curated datasets, such as near and far OOD datasets (Yang et al., 2022).

---

[1]The code is publicly available at https://github.com/natixx14/SPADE

Regarding the robustness to adversarial examples, many attack and defense approaches have been designed along an ongoing arms race; see e.g. (Yan et al., 2018; Zhang et al., 2019; Wang et al., 2020; Hosseini et al., 2020; Huang et al., 2023; Mao et al., 2019; Liu et al., 2023; Carmon et al., 2019; Wang et al., 2023) (more in Section 6). It should be noted that human experts and models tend to make different decisions when handling both OOD and adversarial samples, albeit in distinct ways. Experts typically recognize an OOD sample as belonging to a known class, despite its atypicality, while the model assigns it to a random class. Conversely, experts perceive an adversarial example as typical of a specific class, yet the model confidently misclassifies it into a different class.

The approach presented in this paper, referred to as *Sample-efficient ProbAbilistic Detection using Extreme value theory* (SPADE) and inspired by distance-based approaches (Sun et al., 2022), introduces an original model of the training distribution with respect to a learned model (hereafter the teacher). Specifically, the distances between samples in the teacher latent space are modeled using the Extreme Value Theory (EVT) (Fisher & Tippett, 1928). This model provides a sound and robust test for detecting and rejecting OOD samples; under mild assumptions this test also provably rejects adversarial examples with high probability, subject to a bound on their perturbation amplitude.

The contributions of the proposed approach are fourfold: i) it introduces a formal definition of OOD samples in relation to a teacher and its latent representation; ii) this definition leads to a statistically frugal OOD test based on EVT first principles; iii) this test operationally rejects OOD and adversarial samples with provable guarantees; iv) the effectiveness of the approach is experimentally and successfully validated against strong baselines for teachers with different neural architectures (He et al., 2016; Dosovitskiy et al., 2021; Ming et al., 2023).

The limitation of the approach in detecting adversarial samples lies in its dependence on the Lipschitz constant of the teacher. A relaxed characterization is therefore proposed, leveraging Lipschitz constants that are valid in the local neighborhood of the training samples. Experimental analysis of these local Lipschitz constants confirms the practical usefulness of the approach.

The paper is organized as follows. Section 2 outlines the formal background of distance-based OOD detection and introduces extreme value theory. Section 3 provides an overview of the SPADE approach and presents its formal analysis. Sections 4 and 5 respectively detail the experimental setup and the experiments conducted to validate SPADE against state-of-the-art methods. Section 6 discusses the contributions within the context of related work, and the paper concludes with perspectives for future research.

**Notations.** Let $\mathcal{D} = \{(\mathbf{x}_1, y_1), \ldots, (\mathbf{x}_n, y_n)\}$ denote the training set, iid drawn after the joint distribution $P_{X,Y}$, with $\mathcal{X}$ the instance space and $\mathcal{Y} = \{1, \ldots, n_c\}$ the set of classes. The trained teacher $f$ is expressed as $f = g \circ h$, where $h$ is the embedding in the latent space ($h : \mathcal{X} \mapsto \mathbb{R}^d$) and $g$ is the mapping used to make decisions based on the latent representation.

## 2 FORMAL BACKGROUND

This section describes the main concepts defined by (Ye et al., 2021) in the context of distance-based OOD generalization, and briefly introduces extreme value theory for completeness.

### 2.1 PROPERTIES OF LATENT IN-DISTRIBUTION

As mentioned earlier, the complexity of OOD characterization arises from the highly complex and diverse nature of real-world data (Farquhar & Gal, 2022). In the literature (Cimpoi et al., 2014; Horn et al., 2018; Wang et al., 2022; Vaze et al., 2022; Bitterwolf et al., 2023), OOD characterization often relies on subjective assessment ("data that appear noticeably different from in-distribution to human observers"). A common used definition states that the OOD data are different from the known in-distribution, e.g., the training distribution (hereafter ID). However, this definition does not fully capture the specifics of OOD: OOD is not merely any distribution that differs from ID.

On the other hand, as noted by Papernot & McDaniel (2018); Sun et al. (2022), the latent space representation of in-distribution data often presents distinct characteristics, typically forming compact and well-separated clusters. Ye et al. (2021) formalize these properties in terms of variation and informativeness of the latent representation:

**Definition 1** (Ye et al. (2021))**.** *The* variation of embedding $h$ *across a finite distribution $\mathcal{D}$, noted $\mathcal{V}_\rho(h, \mathcal{D})$, is defined as the maximum diameter over all classes $c$ of the ball containing the distribution $h(\mathbf{x})$ for $(\mathbf{x}, y) \in \mathcal{D}$ and $y = c$:*

$$\mathcal{V}(h, \mathcal{D}) = \max_{c \in Y} \sup_{\substack{(\mathbf{x}, c) \in \mathcal{D} \\ (\mathbf{x}', c) \in \mathcal{D}}} \| h(\mathbf{x}) - h(\mathbf{x}') \| \tag{1}$$

*where the distance $\| h(\mathbf{x}) - h(\mathbf{x}') \|$ is usually set to $L_2$ distance[2]. Embedding $h$ is said $\eta$-invariant across $\mathcal{D}$ if $\mathcal{V}(h, \mathcal{D}) < \eta$.*

The variation of embedding $h$ thus measures the maximum thickness and width of the latent manifold containing the image of (samples in) a class.

**Definition 2** (Ye et al. (2021))**.** *The* informativeness of embedding $h$ *across a finite distribution $\mathcal{D}$, noted $\mathcal{I}(h, \mathcal{D})$, is defined as the average over all pairs $(c, c')$ of distinct classes, of the minimum distance between $h(\mathbf{x})$ and $h(\mathbf{x}')$ for $(\mathbf{x}, y)$ and $(\mathbf{x}', y')$ in $\mathcal{D}$, with $y = c \neq y' = c'$:*

$$\mathcal{I}(h, \mathcal{D}) = \frac{1}{n_c(n_c - 1)} \sum_{c \neq c' \in \mathcal{Y}} \min_{\substack{(\mathbf{x}, y) \in \mathcal{D}, (\mathbf{x}', y') \in \mathcal{D} \\ y = c \neq y' = c'}} \| h(\mathbf{x}) - h(\mathbf{x}') \| \tag{2}$$

*where the distance is usually set to $L_2$ distance (see footnote 1), and $n_c$ denotes the number of classes. The latent embedding $h$ is said $\delta$-informative across $\mathcal{D}$ if $\mathcal{I}(h, \mathcal{D}) > \delta$.*

It is noted that informativeness and variation are closely related to the *compactness* and *dispersion* metrics introduced and optimized in CIDER to achieve OOD generalization (Ming et al., 2023).

These definitions are expressed in terms of the supremum over classes, raising the challenge of their statistical stability and algorithmic exploitation (e.g., setting appropriate thresholds). The approach presented here addresses this challenge by exploiting extreme value theory, as described below, and referring the reader to (Haan & Ferreira, 2006) for a comprehensive introduction.

## 2.2 EXTREME VALUE THEORY

Dating back to Fisher & Tippett (1928); Gnedenko (1943), Extreme Value Theory (EVT) focuses on modeling and understanding the tail behavior of distributions. EVT is based on the premise that, under mild assumptions, the distributions of extreme events converge to a common form, even if their original distributions differ. For instance, while the distributions of seismic intensities and the heights of rogue waves − factors that respectively influence the design of buildings and oil rigs − may differ, their extreme values (maxima) follow the same class of distributions. This limiting distribution is known as the *Extreme Value Distribution* (EVD):

**Definition 3** (Extreme Value Distribution (EVD) (Fisher & Tippett, 1928))**.** *Let $Z$ be a random variable over the real-valued space $\mathbb{R}$. Let $Z^{(\ell)}$ be the random variable defined as the maximum value over $\ell$ independent drawings of $Z$. When $\ell$ goes to infinity, the limiting distribution of $Z^{(\ell)}$ is the cumulative distribution $P(Z^{(\ell)} < z) \underset{\ell \to \infty}{\to} G_{\xi, \mu, \sigma}(z)$, expressed as one of the two parametric models:*

$$G_{\xi, \mu, \sigma}(z) = \exp \left\{ \begin{array}{ll} \left( 1 + \xi \frac{z - \mu}{\sigma} \right)_+^{-1/\xi} & \text{if } \xi \neq 0 \\ -\exp \left( \frac{\mu - z}{\sigma} \right) & \text{otherwise} \end{array} \right\} \tag{3}$$

*with $\mu \in \mathbb{R}$ a location parameter, $\sigma \in \mathbb{R}_+$ a dispersion parameter and $\xi \in \mathbb{R}$ a shape parameter referred to as* extreme value index.

Overall, the EVT framework provides a general parametric model for extreme events associated with a random variable $Z$, independent of the underlying distribution of $Z$ itself. The universality of these models reflects the fact that modeling the extreme events associated with a distribution relies only on the behavior of the tail of the distribution. This tail can take one of three forms: (i) an exponential tail ($\xi = 1$, corresponding to the Gumbel distribution); (ii) a heavy tail ($\xi > 0$, corresponding to the Fréchet distribution); or (iii) a bounded tail ($\xi < 0$, corresponding to the Weibull distribution).

---

[2]The Kullback Leibler divergence is considered when embedding $h$ is a probabilistic one. See Ye et al. (2021) for more detail.

Despite its broad applicability, EVT has, to the best of our knowledge, seen limited use in machine learning, with notable exceptions in the area of anomaly detection (Smith et al., 2012; Siffer et al., 2017; French et al., 2019).

## 3 SPADE OVERVIEW

SPADE is built upon the characterization of the training distribution (hereafter referred to as in-distribution, ID) with respect to the trained teacher model and its associated latent representation. This formal characterization relies on generalized extreme value (GEV) models, which allow for detecting and rejecting out-of-ID samples. For simplicity, and by abuse of language, out-of-ID samples are referred to as OOD in the following. Interestingly, under mild assumptions, the GEV models also allow for detecting adversarial samples. SPADE thus defines an abstaining classifier, that behaves like the teacher on all samples but those that are rejected by the GEV-based detection tests. This abstaining classifier provides probabilistic guarantees of OOD and adversarial sample rejection, subject to a lower bound on the magnitude of the adversarial perturbation.

### 3.1 EVT-BASED CHARACTERIZATION OF OOD

The distance-based OOD detection literature (see, e.g., (Papernot & McDaniel, 2018; Dziedzic et al., 2022; Sun et al., 2022)) suggests that a sample is *likely* to be an OOD sample if it is *sufficiently* distant from the training samples of all (or most) classes in the latent space.

In SPADE, this intuition is formalized using generalized extreme value models, directly yielding the probability for a sample to be OOD. Let $(X, Y)$ denote the random variable following the joint distribution $P_{X,Y}$. For $Y = c$, let $Z_c$ be the random variable defined as the distance between $h(X)$ and its $k$-th nearest neighbor in latent distance, belonging to $\mathcal{D}$ with same class $c$. By definition, the limiting distribution of the maxima of $Z_c$ follows a Generalized Extreme Value model noted $G^{(c)}$, with $Pr(Z_c^{(\ell)} < v) \underset{\ell \to \infty}{\to} G^{(c)}(v)$.

For each sample $\mathbf{x}$ in the instance space and each class $c$, let $z_c$ be defined as $\|h(\mathbf{x}) - h(\mathbf{x}_{knn,c})\|$ where $\mathbf{x}_{knn,c}$ is the $k$-th nearest neighbor of $\mathbf{x}$ in latent space, such that $(x_{knn,c}, c)$ belongs to $\mathcal{D}$. As the true label of $\mathbf{x}$ is unknown at inference time, the proposed OOD test retains the lowest probability of $\mathbf{x}$ being OOD according to all $G^{(c)}$:

**Definition 4** (OOD test). *Let $\mathbf{x}$ denote an instance in $\mathcal{X}$ with $z_c$ denoting its Euclidean latent distance to its k-nearest neighbor of class c in the training set $\mathcal{D}$. The probability of $\mathbf{x}$ being an OOD sample, denoted $OOD(\mathbf{x})$, is defined as:*

$$OOD(\mathbf{x}) = \min_{c \in \mathcal{Y}} G^{(c)}(z_c) \tag{4}$$

In other words, $\mathbf{x}$ is considered OOD if it is extreme according to each and every GEV model associated with one class. The choice to consider a separate GEV $G^{(c)}$ for each class $c$ is aimed at handling cases where classes exhibit different levels of variation in the latent space (Def. 1). In such cases, using a single GEV model for all classes could result in either erroneously rejecting samples from a class with high variation or incorrectly accepting OOD samples from a class with low variation. Since the OOD test depends on the classifier's latent representation, one might wonder to what extent different tests based on different classifiers are consistent; we shall return to this issue in Section 5.

### 3.2 ABSTAINING CLASSIFIER ON OOD SAMPLES

The SPADE abstaining classifier is built as follows:

**Definition 5** (Abstaining classifier). *Given teacher $f$ and confidence $1 - \tau$, with $0 < \tau < 1$, classifier $f_\tau$ abstains from making predictions on a sample $\mathbf{x}$ if $\mathbf{x}$ is considered to be extreme with probability at least $1 - \tau$ w.r.t. its candidate class $c = f(\mathbf{x})$. With same notations as above:*

$$f_\tau(\mathbf{x}) = \begin{cases} f(\mathbf{x}) & \text{if } z_c \leq G^{(c)^{-1}}(1 - \tau) \\ \text{abstain} & \text{otherwise} \end{cases} \tag{5}$$

*where $z_c$ is the distance between $h(\mathbf{x})$ and its nearest neighbor of class c in $\mathcal{D}$.*

Note that the OOD test (Eq. 4) is more tight than the above abstention test (Eq. 5), as the latter exploits the class $f(\mathbf{x})$ predicted by the teacher for $\mathbf{x}$. However, both tests coincide under the common assumption that an OOD sample lies closer to the samples in its class than to samples in other classes, everything else being equal.

### 3.3 ABSTAINING CLASSIFIER WITH PROVABLE GUARANTEES W.R.T. ADVERSARIAL EXAMPLES

By design, the SPADE abstaining classifier $f_\tau$ rejects the samples outside the training distribution with probability $1 - \tau$. Moreover, under the assumption that the latent embedding $h$ admits a finite Lipschitz constant, $f_\tau$ also provably rejects the adversarial samples built by perturbing a training sample, subject to a lower bound on the perturbation amplitude.

Formally, let us consider an adversarial example $\mathbf{x}$ built by perturbing a training sample noted $\mathbf{x}^*$, with a perturbation amplitude $\varepsilon = \|\mathbf{x} - \mathbf{x}^*\|$. Let $c$ be the class of $\mathbf{x}^*$ and let us assume that the class assigned to $\mathbf{x}$ is different from that of $\mathbf{x}^*$: $f(\mathbf{x}) = c' \neq c$.

Let $G^{(c,c')}$ denote the generalized extreme value model associated with the *minimum* latent distance among pairs of examples $(\mathbf{x}, \mathbf{x}')$ respectively belonging to class $c$ and $c'$ (akin the informativeness, Def. 2):

$$G^{(c,c')}(v) = Pr\Big(\|h(X) - h(X')\| > v \mid (X, Y) \sim P_{X,Y}, \ (X', Y') \sim P_{X,Y}, \ Y = c, \ Y' = c'\Big)$$

**Theorem 1** (Global Pertubation Bound). *Assume that the latent embedding $h$ is $K$-Lipschitz. Let $\mathbf{x}$ be an adversarial sample obtained by perturbing a training sample $\mathbf{x}^*$ of class $c$, with perturbation amplitude $\varepsilon = \|\mathbf{x} - \mathbf{x}^*\|$, and let $f(\mathbf{x}) = c' \neq c$. Let $\mathbf{x}'^*$ of class $c'$ denote the $k$-th nearest training sample in $\mathcal{D}$ of $\mathbf{x}$. Then, with probability at least $1 - \tau$ either $f_\tau$ abstains on $\mathbf{x}$, or $\varepsilon$ admits the following lower bound:*

$$\varepsilon \geq \frac{1}{K}\left(G^{(c,c')^{-1}}(1 - \tau) - G^{(c')^{-1}}(1 - \tau)\right) \tag{6}$$

*Proof.* If $\|h(\mathbf{x}) - h(\mathbf{x}'^*)\| > (G^{(c')})^{-1}(1 - \tau)$, then $f_\tau$ abstains on $\mathbf{x}$ after Eq. 5. Otherwise,

$$\|h(\mathbf{x}^*) - h(\mathbf{x}'^*)\| \leq \|h(\mathbf{x}^*) - h(\mathbf{x})\| + \|h(\mathbf{x}) - h(\mathbf{x}'^*)\|$$
$$\leq K\varepsilon + (G^{(c')})^{-1}(1 - \tau) \tag{7}$$

Moreover, with probability $1 - \tau$ after the definition of $G^{(c,c')}$,

$$(G^{(c,c')})^{-1}(1 - \tau) \leq \|h(\mathbf{x}^*) - h(\mathbf{x}'^*)\| \tag{8}$$

Putting together Eqs. 7 and 8 concludes the proof. $\square$

A main limitation of the above result is that the lower bound on $\varepsilon$ becomes trivial if embedding $h$ is associated with a very high Lipschitz constant. This limitation is alleviated by considering instead local Lipschitz constants, taking inspiration from (Jordan & Dimakis, 2020; Shi et al., 2022), and providing local guarantees.

**Theorem 2** (Local Perturbation Bound). *Let $\mathbf{x}^*$ be a training sample of class $c$. Let $r$ be a positive real value, and let us assume that embedding $h$ admits a local Lipschitz constant $K_r(\mathbf{x}^*)$ in the vicinity of $\mathbf{x}^*$:*

$$\forall \mathbf{x} \in \mathcal{X} \ s.t. \ \|\mathbf{x} - \mathbf{x}^*\| < r, \|h(\mathbf{x}) - h(\mathbf{x}^*\| < K_r(\mathbf{x}^*)\|\mathbf{x} - \mathbf{x}^*\|$$

*Let $\mathbf{x}$ be an adversarial sample built by perturbation of $\mathbf{x}^*$ such that $\varepsilon = \|\mathbf{x} - \mathbf{x}^*\| < r$, with $f(\mathbf{x}) = c' \neq c$.*
*Then, with probability at least $1 - \tau$, either $f_\tau$ abstains on $\mathbf{x}$, or $\varepsilon$ satisfies the following lower bound:*

$$\varepsilon \geq \frac{1}{K_r(\mathbf{x}^*)}\left(G^{(c,c')^{-1}}(1 - \tau) - G^{(c')^{-1}}(1 - \tau)\right) \tag{9}$$

*Proof.* Same as proof of Thm 1. $\square$

The guarantee thus depends on the local Lipschitz constants associated with the training samples. The range of these constants is presented in Supplementary Material (SM, section E).

## 3.4 ESTIMATING THE GEV MODELS

The estimation of the GEV models used in SPADE is detailed in the case of $G^{(c)}$, defining its approximation $\widehat{G}^{(c)}$. The same process is applied to learn an approximation of the $G^{(c,c')}$ models, with the only difference being that the considered extreme values are minima instead of maxima.

After Siffer et al. (2017), a straightforward approximation of an EVD proceeds by sampling the extreme events along the block maxima method. However, This fitting process is found to be sensitive to both the number of blocks and the block size. The proposed approach thus is the *Peak Over Threshold* (POT) method, which relies on the Pickands-Balkema-de-Haan theorem, often referred to as the *second theorem of EVT* (Balkema & de Haan, 1974; Pickands, 1975). Formally, the POT considers the occurrences of events bypassing a given threshold $t$, noting that the distribution of these occurrences follows a Generalized Pareto Distribution (GPD):

$$F_{\xi,\mu,\sigma}(z) = Pr(Z - t > z | Z > t) = \begin{cases} 1 - \exp\left(-\frac{z-\mu}{\sigma}\right) & \text{if } \xi = 0 \\ 1 - \left(1 + \frac{\xi(z-\mu)}{\sigma}\right)^{-1/\xi} & \text{otherwise} \end{cases} \quad (10)$$

Informally, for each class $c$, POT proceeds by fitting a GPD model to samples the distances $z_c$ that exceed a predefined threshold $t_c$. Formally, for each sample $(\mathbf{x}_i, c)$ in $\mathcal{D}$, let $z_i \in \mathbb{R}$ be the distance between $\mathbf{x}_i$ and its $k$-th nearest neighbor belonging to class $c$ in latent space. Let $\mathcal{D}_c$ be the set including all such $z_i$ for $z_i > t_c$.

The parameters $(\mu_c, \sigma_c, \xi_c)$ of model $\widehat{G}^{(c)}$ are learned by maximum likelihood estimation (MLE) on $\mathcal{D}_c$:

$$(\mu_c, \sigma_c, \xi_c) = \arg\max_{\mu,\sigma,\xi} \sum_{z \in \mathcal{D}_c} \mathcal{L}_{\mu,\sigma,\xi}(z) \quad (11)$$

with $\mathcal{L}$ the log-probability density function of the GPD. MLE is preferred over alternative methods, such as the method of moments, due to its higher robustness and efficiency (Siffer et al., 2017).

Eventually, model $\widehat{G}^{(c)}$ approximately characterizes whether a given sample is OOD with respect to class $c$ with confidence $1 - \tau$. The OOD test (Def. 4) is accordingly approximated as:

$$\widehat{OOD}(\mathbf{x}) = \min_{c \in \mathcal{Y}} \widehat{G}^{(c)}(z_c) \quad (12)$$

Overall, the SPADE abstaining classifier relies on Def. 5, where the GEV models are learned using Alg. 1.

---

**Algorithm 1** SPADE. Learning EVT models

---

1: **Input:** training set $\mathcal{D}$, integer $k$, threshold $t > 0$.
2: **Output:** GEV distributions $(\widehat{G}^{(c)})$ of each class $c$.
3:
4: **for** $c \in \mathcal{Y}$ **do**
5:    $\mathcal{T}^{(c)} = \{\}$
6:    **for** $(\mathbf{x}, c) \in \mathcal{D}$ **do**
7:       Compute normalized activation: $\mathbf{z} = h(\mathbf{x})/\|h(\mathbf{x})\|$
8:       Compute distance to $k$-th nearest neighbor in class $c$: $v = \min_{(\mathbf{x}',c) \in \mathcal{D}}^{(k)} (\|\mathbf{z} - \mathbf{z}'\|)$
9:       **if** $v > t$: $\mathcal{T}^{(c)} = \mathcal{T}^{(c)} \cup \{v\}$ **end if**
10:    **end for**
11:    Fit an Extreme Value Distribution $(\widehat{G}^{(c)})$ from extreme samples of $\mathcal{T}^{(c)}$
12: **end for**
13: **return** $(\widehat{G}^{(c)})_{c \in \mathcal{Y}}$

---

## 3.5 DISCUSSION

SPADE retains the main benefits of distance-based OOD detection approaches, as it is agnostic to the structure of the OOD distribution and easy to implement. Furthermore, being grounded in

the first EVT principles, it enables estimating the probability for a sample to be OOD. Lastly, the abstaining classifier $f_\tau$ also rejects adversarial samples, subject to a lower bound on the magnitude of adversarial perturbations.

A potential weakness is the complexity of approximating GEV models, quadratic with respect to the number $n$ of samples. This raises the question of whether stable and accurate GEV approximations can be achieved when aggressively subsampling the training set. A second issue concerns the effectiveness of the lower bound used to reject adversarial samples (Eq. 6); specifically the question is whether $G^{(c,c')^{-1}}(1-\tau) - G^{(c)^{-1}}(1-\tau)$ is strictly positive for non-trivial confidence levels $1-\tau$.[3] Both questions will be examined experimentally.

## 4 EXPERIMENTAL SETTING

This section outlines the experimental setup used to evaluate SPADE in comparison to state-of-the-art methods. All experiments are conducted on Tesla A100 80GB GPUs. Additional details are provided in the Supplementary Material (SM).

**Goals.** The experiments aim to empirically answer four questions: the accuracy of the SPADE OOD detection tests, particularly in comparison with distance-based approaches (Q1); their sensitivity with respect to the considered teacher (Q2); similarly, their accuracy of the adversarial sample detection tests, compared with state-of-the-art methods (Q3); the computational complexity and stability of the approximate GEV models embedded in SPADE (Q4).

The performance of SPADE is compared to that of five established baselines: the seminal MSP (Hendrycks & Gimpel, 2017), ODIN (Liang et al., 2018), MDS (Lee et al., 2018), KNN (Sun et al., 2022), and its extension CIDER (Ming et al., 2023) (further discussed in Section 6).

**Metrics.** The comparative evaluation is conducted using the OpenOOD framework (Zhang et al., 2023), with performance assessed by standard indicators: The *Area Under the ROC Curve (AUC)* measures the average rate of correct OOD/adversarial sample detection across all confidence levels, corresponding to the true positive rate as described in (Zhang et al., 2023). The *FPR95 indicator* represents the fraction of true samples misclassified as OOD (respectively, adversarial) when the detection threshold ensures 95% of OOD (resp., adversarial) samples are correctly rejected.

**Benchmarks.** Three medium- and large-sized datasets are considered: CIFAR-10, CIFAR-100 (Krizhevsky, 2009) and ImageNet-1K (using the ILSVRC2012 version). Three types of neural architectures are used to assess the generality of the SPADE approach: ResNet (He et al., 2016), ViT (Dosovitskiy et al., 2021), and VGG (Simonyan & Zisserman, 2015). The distance in latent space between a sample and its $k$-th nearest neighbor is calculated using the normalized $L_2$ distance, following Sun et al. (2022). OOD samples are sourced from near-OOD and far-OOD datasets, following (Cimpoi et al., 2014; Vaze et al., 2022; Horn et al., 2018; Wang et al., 2022; Bitterwolf et al., 2023), as detailed in SM. Adversarial samples are generated by perturbing training samples using FGSM (Goodfellow et al., 2015), PGD (Madry et al., 2018) and AutoAttack (Croce & Hein, 2020) attacks, with a perturbation amplitude $\varepsilon$ ranging from 0.001 to 0.004.

## 5 EXPERIMENTAL RESULTS

**OOD Detection (Q1).** The performance of SPADE is reported in Table 1, for a ResNet teacher in the representative case of ImageNet-1K. For the considered near-OOD datasets, the best-performing method is MSP, while for far-OOD datasets, KNN achieves the best results. In all cases except one, SPADE-ResNet is slightly outperformed by KNN. In terms of ranking (determined by the average of AUC and FPR95), SPADE ranks second best on both near- and far-OOD datasets.

**Sensitivity of OOD Detection (Q2).** The impact of the considered teacher on the performance of the SPADE OOD test is illustrated in Table 2, comparing SPADE built on teachers ResNet, VGG,

---

[3]Note that the requirement for $(G^{(c,c')})^{-1}(1-\tau)$ to be sufficiently large is reminiscent of the clustering assumption that underpins semi-supervised learning (Rigollet, 2007).

Table 1: OOD detection on ImageNet-1K: performance of SPADE with ResNet teacher compared to that of baselines MSP, ODIN, MDS and KNN in terms of AUC (the greater the better) and FPR95 (the lower the better; best performances in bold). The rank, averaged over far and near OOD datasets, is computed based on the half sum of AUC and FPR95.

| | Near OOD | | | | Far OOD | | | | | | Rank |
| | SSB Hard | | NINCO | | iNaturalist | | Textures | | OpenImages-O | | |
| | AUC ↑ | FPR95 ↓ | AUC ↑ | FPR95 ↓ | AUC ↑ | FPR95 ↓ | AUC ↑ | FPR95 ↓ | AUC ↑ | FPR95 ↓ | |
|---|---|---|---|---|---|---|---|---|---|---|---|
| MSP (1) | **72.53** | **74.43** | **80.66** | **57.72** | 87.78 | 44.08 | 82.81 | 59.16 | 85.21 | 49.62 | 3 |
| ODIN (2) | 72.51 | 77.36 | 77.55 | 70.83 | **89.51** | **41.46** | 87.02 | 56.58 | 86.33 | 54.10 | 4 |
| MDS (3) | 52.15 | 90.46 | 68.49 | 71.66 | 76.49 | 56.07 | 94.11 | 27.07 | 77.68 | 59.66 | 5 |
| KNN (4) | 62.80 | 84.08 | 79.30 | 58.92 | 84.62 | 42.39 | **96.06** | **23.39** | **86.38** | **44.24** | 1 |
| **SPADE** | 61.91 | 85.27 | 77.99 | 61.04 | 85.26 | 44.84 | 95.86 | 24.63 | 85.79 | 46.33 | 2 |

Table 2: OOD detection on CIFAR-10: performance of SPADE with teachers ResNet, VGG and ViT-B16, compared to that of baseline CIDER.

| | TIN | | MNIST | | SVHN | | Textures | | Places-365 | |
| | AUC ↑ | FPR95 ↓ | AUC ↑ | FPR95 ↓ | AUC ↑ | FPR95 ↓ | AUC ↑ | FPR95 ↓ | AUC ↑ | FPR95 ↓ |
|---|---|---|---|---|---|---|---|---|---|---|
| CIDER ($d=512$) | 71.56 | 70.34 | 68.84 | 71.86 | 57.51 | 78.43 | 71.06 | 70.70 | 71.73 | 69.97 |
| ResNet-18 ($d=512$) | 90.41 | 32.14 | 93.646 | 21.91 | 92.17 | 22.94 | 91.97 | 25.68 | 91.03 | 30.41 |
| VGG-16 ($d=512$) | 76.04 | 66.84 | 89.37 | 35.12 | 81.53 | 44.06 | 80.47 | 50.73 | 72.14 | 77.27 |
| ViT-B16 ($d=384$) | 96.27 | 20.18 | 94.52 | 12.23 | 81.78 | 34.83 | 99.97 | 00.06 | 99.67 | 00.51 |

ViT-B16 with the CIDER baseline on CIFAR-10. These results show that the detection accuracy indeed depends on the teacher and its latent space, with SPADE-ViT-B16 dominating ResNet (except on SVHN) and ResNet strongly dominating VGG.

Still, the performance does not merely depend on the size of the latent space. The discrepancy between the AUC and FPR95 indicators suggests that the optimal detection threshold varies depending on the dataset. Notably, all SPADE OOD detection tests outperform CIDER, which is based on KNN and specifically targets OOD detection. We shall come back to this remark in Section 6.

**Detection of Adversarial Samples (Q3).** Table 3 reports the performance of SPADE with a ResNet teacher on the representative cases of CIFAR-10 and CIFAR-100, compared with MSP, MDS, KNN and CIDER. For all perturbation amplitudes, SPADE ranks first w.r.t. AUC (and second w.r.t. FPR95). In terms of FPR95, KNN ranks first on CIFAR-10, while MSP ranks first on CIFAR-100. Overall, SPADE behaves on par with, or better than OOD detection methods w.r.t. the detection of adversarial examples. The slight AUC improvement suggests that SPADE may capture more subtle differences between in-distribution and adversarial samples. Conversely, the high FPR95 values suggest that SPADE tends to be overly cautious, rejecting true samples at the confidence level where 95% adversarial samples are rejected.

**Computational Frugality: Stability of GEV Models and SPADE Performances (Q4).** The stability of the GEV models is analyzed by sub-sampling the training set. Fig. 1

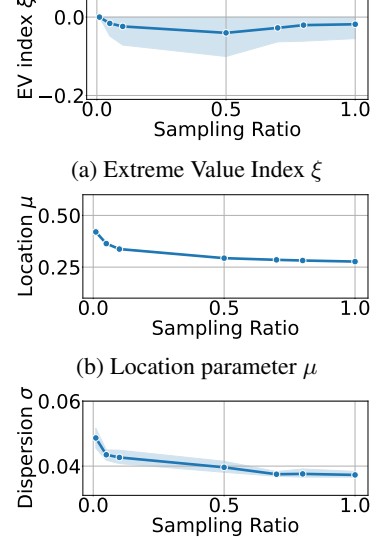

(a) Extreme Value Index $\xi$

(b) Location parameter $\mu$

(c) Digression parameter $\sigma$

Figure 1: Stability of EVT parameter estimation wrt sampling ratio and estimation variance for one class of CIFAR-100 on ResNet-18.

reports the $\widehat{G}^{(c)}$ hyperparameters *vs* the sub-sampling rate, in the case of the ResNet teacher on CIFAR-100. The figure shows: i) a low sensitivity of the tail index (Fig. 1.(a)); ii) a decrease in $\mu$ with the sub-sampling rate (Fig. 1.(b)); iii) a low sensitivity of the dispersion parameter $\sigma$ (Fig. 1.(c)). Additional results are presented in SM, confirming the relevance of the lower bound on the amplitude of the adversarial perturbation (Eqs. 6 and 9).

Table 3: Adversarial samples detection on CIFAR-10 and CIFAR-100, with perturbation amplitude from .001 to .004: comparison of SPADE (ResNet teacher) with baselines MSP, ODIN, MDS and KNN in terms of AUC (the greater the better) and FPR95 (the lower the better; best in bold).

| | | $\varepsilon = 0.001$ | | $\varepsilon = 0.002$ | | $\varepsilon = 0.003$ | | $\varepsilon = 0.004$ | | Average | |
| --- | --- | --- | --- | --- | --- | --- | --- | --- | --- | --- | --- |
| | | AUC ↑ | FPR95 ↓ | AUC ↑ | FPR95 ↓ | AUC ↑ | FPR95 ↓ | AUC ↑ | FPR95 ↓ | AUC ↑ | FPR95 ↓ |
| CIFAR-10 | MSP (1) | 81.68 | 79.08 | 81.90 | 78.70 | 82.06 | 79.14 | 82.20 | 78.10 | 81.96 | 78.76 |
| | MDS (3) | 81.46 | 69.34 | 81.51 | 68.76 | 81.57 | 69.04 | 81.61 | 69.42 | 81.54 | 69.14 |
| | KNN (4) | 85.65 | **54.46** | 85.75 | **54.19** | 85.85 | **53.91** | 85.92 | 54.29 | 85.79 | **54.21** |
| | CIDER (5) | 85.46 | 55.68 | 85.50 | 54.79 | 85.53 | 54.93 | 85.60 | 54.57 | 85.52 | 54.99 |
| | **SPADE** | **85.96** | 55.02 | **86.06** | 54.51 | **86.15** | 54.40 | **86.24** | **53.78** | **86.10** | 54.43 |
| CIFAR-100 | MSP (1) | 83.24 | **51.84** | 83.39 | **50.57** | 83.54 | **49.94** | 83.68 | **49.64** | 83.46 | **50.50** |
| | MDS (3) | 60.34 | 82.93 | 60.29 | 83.01 | 60.25 | 82.60 | 60.19 | 83.01 | 60.27 | 82.89 |
| | KNN (4) | 83.54 | 56.83 | 83.67 | 56.39 | 83.79 | 55.64 | 83.89 | 55.13 | 83.72 | 56.00 |
| | CIDER (5) | 82.75 | 63.17 | 82.85 | 63.57 | 82.95 | 63.17 | 83.06 | 62.34 | 82.90 | 63.06 |
| | **SPADE** | **84.33** | 53.13 | **84.45** | 52.84 | **84.56** | 52.44 | **84.66** | 52.44 | **84.50** | 52.72 |

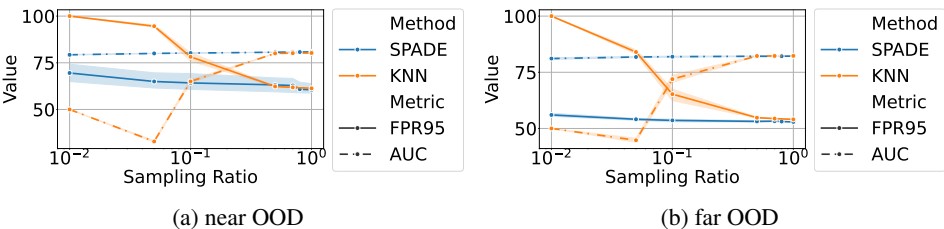

(a) near OOD  (b) far OOD

Figure 2: Sensitivity analysis of the OOD detection on CIFAR-100 w.r.t. the subsampling rate of the training set: AUC (dashed line) and FPR95 (solid line) performances for SPADE (in blue) and KNN (Sun et al., 2022) (in orange; better seen in color).

As expected, the stability of the $\widehat{G}^{(c)}$ models results in stable OOD detection performances (AUC and FPR95) when subsampling the training set (Fig. 2). Quite the contrary, the OOD detection performances of KNN are significantly deteriorated when aggressively subsampling the training set, particularly so on near-OOD (Fig. 2, left).

## 6 POSITION W.R.T. RELATED WORK

As said, the lack of robustness of ML models w.r.t. OOD and adversarial samples remains a major obstacle for deploying them in real-world settings (Salehi et al., 2022; Yang et al., 2024). Consequently, a hot research topic is concerned with designing more robust learning methods.

The detection of OOD samples has been tackled along different approaches, ranging from learning with rejection (Bartlett & Wegkamp, 2008), to anomaly detection (Bulusu et al., 2020), novelty detection (Markou & Singh, 2003a;b), and open set recognition (Boult et al., 2019). To the best of our knowledge, the OOD detection problem was first formalized by Nguyen et al. (2015); the early MSP approach, observing the margin between the logits of the trained teacher and exploiting the fact that it behaves differently for in-distribution and OOD samples (Hendrycks & Gimpel, 2017), still is among the most effective ones (Table 1). Along the same line, ODIN exploits the gradient information to separate in- and out-of-distribution samples (Liang et al., 2018).
Some approaches address OOD detection as yet another supervised learning problem, handling OOD samples as if they belonged to an additional class and (re)-training the model accordingly (Du et al., 2022a; Zhang et al., 2022) (see also (Hsu et al., 2020)).

Quite a few other methods, based on the so-called manifold assumption, focus on identifying the manifold on which real samples lie; one option is to consider the latent representation of an auto-encoder (AE) trained solely on real samples. As empirically shown by e.g. Ren et al. (2019); Liu et al. (2021); Du et al. (2022a), the reconstruction error of OOD samples through the AE is much higher than for real samples; an effective OOD test can thus be based on this error. On the positive side, this reconstruction error defines a general criterion and does not depend on the teacher; on

the negative side, it does not leverage class information. Another option is based on modeling the behavior of true samples in the penultimate layer of the neural net, using probabilistic models (Du et al., 2022b; Liu et al., 2021; Ren et al., 2019).

Finally, the option most closely related to SPADE is to consider the latent representation of the teacher itself, as done in distance-based OOD detection approaches (Papernot & McDaniel, 2018; Lee et al., 2018), particularly in KNN (Sun et al., 2022). The difference is that KNN exploits the distance $z$ of a sample to its $k$-th nearest neighbor in the training set, whereas SPADE exploits GEV models and uses $\widehat{G}^{(c)}(z_c)$ to estimate the probability for the sample to be OOD. A possible interpretation for the better performance of KNN (Table 1) is the better bias-variance trade-off in the empirical test based on $z$ compared to the test based on $\widehat{G}^{(c)}(z_c)$. While learning a parametric GEV model achieves some regularization, this model is only trained from examples in class $c$, thus operating with one or two orders of magnitude less data.

The problem of dealing with adversarial examples differs from that of detecting OOD examples, as adversarial examples are deliberately crafted to deceive the teacher (Szegedy et al., 2014; Goodfellow et al., 2015; Madry et al., 2018; Croce & Hein, 2020). Knowing their structure allows for the design of specific defense strategies, such as adversarial training (Goodfellow et al., 2015; Madry et al., 2018), which incorporates adversarial examples into the training process (Yan et al., 2018; Zhang et al., 2019; Wang et al., 2020). Other notable defense strategies include adversarial architectures (Hosseini et al., 2020; Huang et al., 2023), adversarial regularization (Mao et al., 2019; Liu et al., 2023), and data augmentation methods (Carmon et al., 2019; Wang et al., 2023).

In contrast, SPADE neither requires additional information nor retrains the classifier to defend itself against adversarial examples. It employs the same agnostic strategy for both adversarial and OOD threats: it characterizes the true samples. It abstains from deciding on any sample that appears to be extreme compared to the true (ID) samples, according to the chosen confidence level.

## 7 CONCLUSION AND PERSPECTIVES

The main contribution of the paper, SPADE (*Sample-efficient ProbAbilistic Detection using Extreme value theory*), is a statistical test designed to detect samples appearing to be extreme w.r.t. training samples, enabling the classifier to abstain from making decisions on such extreme examples. The ability of this test to accurately detect OOD and adversarial samples has been empirically demonstrated, behaving on par with, or better than, the prominent state of the art approaches.

Like distance-based OOD detection approaches, this test exploits the latent distance between the given example and its nearest neighbor in the training set. Its originality lies in leveraging Extreme Value Theory (Fisher & Tippett, 1928) to provide a formal characterization of the in-distribution samples. This characterization offers two key benefits: first, it yields provable guarantees for detecting adversarial examples, subject to the adversarial perturbation amplitude to be lower bounded; second, it provides some new hints into the key aspects governing the teacher robustness.[4] Lastly, the computational frugality and the stability of the generalized EV models underlying the test have been empirically established.

This approach opens several perspectives, aimed to making classifiers more robust and better understanding the key robustness factors. A short term perspective is to apply the generalized extreme value models to some of the criteria empirically used in the OOD detection literature; one such criterion is the score margin involved in MSP (Hendrycks & Gimpel, 2017). Another perspective is to enhance the classifier training loss to favor the robustness of the latent space, e.g. to consider the optimization of the Lipschitz constant of the classifier embedding besides its variation and informativeness as done in CIDER (Ming et al., 2023).

Our long-term goal is to investigate whether *safe* example behaviors can be identified in the latent space and whether these behaviors can be certified, as a step toward the certification of neural networks.

---

[4]For instance, CIDER (Ming et al., 2023) involves the optimization of the variation and informativeness of the teacher latent space. The SPADE analysis suggests that, beyond these two factors, the regularity of the teacher − its Lipschitz constant − also matters. A possible interpretation for why SPADE outperforms CIDER is that optimizing variation and informativeness might adversely affect this Lipschitz constant.

ACKNOWLEDGMENTS

The authors want to thank Cyril Furtlehner, TAU-INRIA, for many discussions about extreme value theory. We also thank the anonymous reviewers for their critiques, that helped us to improve the paper.

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

SUPPLEMENTARY MATERIAL

The supplementary material is organized as follows:

- Section A presents additional results, including SPADE's accuracy in detecting hard OOD samples (Fort, 2022) and adversarial examples generated using PGD (Madry et al., 2018) and AutoAttack (Croce & Hein, 2020);

- Section B presents a new theoretical result, concerning the detection of adversarial OOD samples (informed attacks);

- Section C provides a more comprehensive introduction to Extreme Value Theory.

- Section D details the experimental setting to ensure reproducibility, specifically describing the benchmark implementations and the hyperparameter configurations.

- Section E investigates the feasibility of local Theorem 2.

## A    COMPLEMENTARY RESULTS

### A.1    OOD DETECTION, ROC CURVE

The Receiver Operating Characteristic (ROC) curve, a standard indicator for evaluating the performance of a continuous model ($h : \mathcal{X} \mapsto \mathcal{R}$), measures the trade-off between the true positive rate and the false positive rate of the binary classifier defined as $g(x) = (h(x) > t)$ for threshold $t$ varying in $\mathcal{R}$. Fig. 3 illustrates the ROC curves for OOD detection on CIFAR-10 and CIFAR-100, comparing the performances of SPADE and KNN. These results confirm SPADE's robustness in detecting OOD and adversary examples, with the same performance as KNN.

### A.2    DETECTION OF NEAR-OOD, FAR-OOD AND ADVERSARIAL EXAMPLES (FGSM)

Table 4 presents the detection performance of OOD and adversarial examples, comparing SPADE with four baseline methods on CIFAR-10 and CIFAR-100 using two performance indicators: AUC (higher is better) and FPR95 (lower is better). On CIFAR-10, SPADE ranks first in 4 out of 6 cases, and second in the remaining cases, behind KNN. On CIFAR-100, SPADE ranks first in 3 out of 6 cases and second two times, behind KNN or MSP. These results highlight the robustness of SPADE in detecting both OOD and adversarial examples.

### A.3    DETECTION OF HARD NEAR-OOD EXAMPLES

As shown by Fort et al. (2021), a more challenging near-OOD detection task consists in distinguishing between CIFAR-10 and CIFAR-100. Table 5 presents the accuracy of SPADE against MSP, MDS, KNN and CIDER, using CIFAR10 as in-distribution vs CIFAR10 as Out-Of-Distribution, and vice versa. Results are reported for two teacher models: VGG (Table 5, left) and ViT (Table 5, right).

For in-distribution CIFAR-10, SPADE ranks first in 2 out of 4 cases and second in the remaining cases, following MSP. For in-distribution CIFAR-100, SPADE ranks second in 3 out of 4 cases, behind either MSP or MDS. These results confirm the general robustness of SPADE.

### A.4    DETECTION OF MORE CHALLENGING ADVERSARIAL EXAMPLES

Complementary results are provided in Tables 6 and 7, which respectively evaluate adversarial examples generated by PGD (Madry et al., 2018) and AutoAttack (Croce & Hein, 2020), for an adversarial perturbation amplitude ranging in $\{10^{-4}, \ldots, 10^{-1}\}$. The comparison of SPADE (with ResNet teacher) against the baselines shows that for PGM attacks (Table 6), SPADE ranks first in 7 out of 8 cases on CIFAR-10 (second otherwise) while it ranks first 5 out of 8 cases on CIFAR-100.

Table 7 further shows that for AA attacks, SPADE ranks first 2 out of 8 cases on CIFAR-10, and first 5 out of 8 cases on CIFAR-100. These results similarly confirm the robustness of SPADE. While SPADE is second behind CIDER in 6 out of 8 cases on CIFAR-10, CIDER generally performs

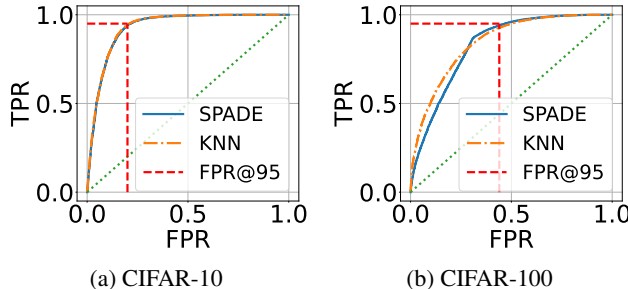

(a) CIFAR-10  (b) CIFAR-100

Figure 3: OOD detection: The ROC curves compare the performance of SPADE and KNN (Sun et al., 2022). Left: in-distribution CIFAR-10. Right: in-distribution CIFAR-100.

Table 4: Comparative performance of SPADE (with teacher ResNet) against MSP, ODIN, MDS and KNN: detection of near OOD, far OOD and adversarial examples on CIFAR-100 and CIFAR-10 using AUROC (↑) and FPR95 (↓) indicators. Statistically significantly best performances are in bold.

|  |  | Near OOD | | Far OOD | | Adversarial | |
|---|---|---|---|---|---|---|---|
|  |  | AUC ↑ | FPR95 ↓ | AUC ↑ | FPR95 ↓ | AUC ↑ | FPR95 ↓ |
| CIFAR-10 | MSP (1) | 87.68 | 53.54 | 91.00 | 31.43 | 81.96 | 78.76 |
| | ODIN (2) | 80.25 | 84.49 | 87.21 | 60.89 | - | - |
| | MDS (3) | 86.72 | 46.23 | 90.20 | 30.31 | 81.54 | 69.14 |
| | KNN (4) | 90.70 | **34.54** | 93.11 | 23.88 | 85.79 | **54.21** |
| | **SPADE** | **90.85** | 34.74 | **93.44** | **22.97** | **86.10** | 54.43 |
| CIFAR-100 | MSP (1) | 80.14 | **55.23** | 77.34 | 59.77 | 83.46 | **50.50** |
| | ODIN (2) | 79.86 | 58.03 | 78.98 | 59.34 | - | - |
| | MDS (3) | 58.68 | 84.22 | 70.65 | 72.04 | 60.27 | 82.89 |
| | KNN (4) | 80.32 | 61.33 | **82.64** | 53.47 | 83.72 | 56.00 |
| | **SPADE** | **80.70** | 59.36 | 81.95 | **53.27** | **84.50** | 52.72 |

slightly worse on CIFAR-100. Similarly, while SPADE ranks second behind MSP in 3 out of 8 cases on CIFAR-100, MSP is generally less effective on CIFAR-10.

## A.5 Detection of Adaptative Attacks

The PGD (Madry et al., 2018) loss has been modified and adapted to inform the attacker of the detection method by augmenting the PGD loss with the distance between $x'$ and its nearest neighbor (to be minimized). This attack, referred to as PGD-ad, generates more challenging adversarial samples for our method, as it primarily targets the detection mechanism of SPADE. A performance comparaison of distance-based techniques is presented in Table 8. It highlights that, for a perturbation amplitude of $10^{-2}$, MDS suffers from an AUC loss of .1 (from 82.96 to 82.86), KNN incurs an AUC loss of .7 (from 86.75 to 86.06), and SPADE suffers from an AUC loss of .2 (from 86.91 to 86.70). For a larger perturbation amplitude of 0.1, MDS suffers an AUC loss of 1, KNN drops by 1.2 and SPADE SPADE decreases by 0.8 while still maintaining the highest ranking. For smaller perturbation amplitudes, the interaction between the two terms of PGD-ad is less straightforward to interpret.

## B Detection of adversarial OOD examples

As shown in Fort et al. (2021); Fort (2022), while latent distance metrics metrics can effectively detect even hard OOD examples, their performance can be significantly degraded by white-box attacks.

Table 5: Detection of hard-OOD examples with VGG teacher (Left) and ViT teacher (Right), for in-distribution CIFAR-10 vs OOD CIFAR-100, and vice-versa. Comparative performances of SPADE against baselines MSP, MDS, CIDER and KNN.

(a) VGG teacher

| In-Distribution | CIFAR-10 | | CIFAR-100 | |
|---|---|---|---|---|
| | AUC ↑ | FPR95 ↓ | AUC ↑ | FPR95 ↓ |
| MSP (1) | **78.63** | 59.90 | 66.89 | **73.98** |
| MDS (3) | 66.96 | 90.69 | 43.78 | 95.63 |
| KNN (4) | 78.01 | 59.53 | 60.47 | 77.41 |
| CIDER (5) | 60.34 | 90.03 | **60.09** | 89.87 |
| **SPADE** | 78.41 | **59.34** | 61.86 | 79.97 |

(b) ViT teacher

| In-Distribution | CIFAR-10 | | CIFAR-100 | |
|---|---|---|---|---|
| | AUC ↑ | FPR95 ↓ | AUC ↑ | FPR95 ↓ |
| MSP (1) | 92.01 | **27.66** | 76.50 | 70.68 |
| MDS (3) | 87.08 | 53.37 | **78.02** | **66.28** |
| KNN (4) | 91.52 | 35.83 | 69.80 | 85.18 |
| CIDER (5) | 60.34 | 90.03 | 60.09 | 89.87 |
| **SPADE** | **92.58** | 32.53 | 77.32 | 71.69 |

Table 6: Adversarial samples detection against PGD (Madry et al., 2018) attacks on CIFAR-10 and CIFAR-100, with perturbation amplitude from $10^{-4}$ to $10^{-1}$: comparison of SPADE (ResNet teacher) with baselines MSP, MDS, CIDER and KNN in terms of AUC (higher is better) and FPR95 (lower is better). Statistically significant best results are highlighted in bold.

| | | $\varepsilon = 0.0001$ | | $\varepsilon = 0.001$ | | $\varepsilon = 0.01$ | | $\varepsilon = 0.1$ | | Average | |
|---|---|---|---|---|---|---|---|---|---|---|---|
| | | AUC ↑ | FPR95 ↓ | AUC ↑ | FPR95 ↓ | AUC ↑ | FPR95 ↓ | AUC ↑ | FPR95 ↓ | AUC ↑ | FPR95 ↓ |
| CIFAR-10 | MSP (1) | 81.50 | 79.60 | 81.69 | 79.10 | 83.19 | 73.58 | 88.82 | 46.44 | 83.8 | 69.68 |
| | MDS (3) | 70.44 | 81.44 | 81.47 | 69.63 | 82.96 | 62.19 | 90.36 | 35.34 | 81.31 | 62.15 |
| | KNN (4) | 85.43 | 56.52 | 85.52 | 55.89 | 86.75 | 52.24 | 92.35 | 28.52 | 87.51 | 48.29 |
| | CIDER (5) | 85.41 | 56.30 | 85.47 | 55.51 | 86.48 | **49.13** | 91.53 | 31.36 | 87.22 | 48.08 |
| | **SPADE** | **85.67** | **56.02** | **85.66** | **54.50** | **86.91** | 50.89 | **92.49** | **27.94** | **87.68** | **47.34** |
| CIFAR-100 | MSP (1) | 83.11 | **52.49** | 83.25 | **51.38** | 84.36 | **48.98** | 86.43 | 42.72 | 84.29 | **48.89** |
| | MDS (3) | 82.69 | 60.38 | 82.82 | 60.34 | 82.77 | 59.95 | 78.51 | 64.49 | 81.70 | 61.29 |
| | KNN (4) | 83.93 | 58.32 | 84.04 | 57.78 | 85.12 | 53.14 | 88.77 | 40.73 | 85.47 | 52.50 |
| | CIDER (5) | 82.68 | 63.49 | 82.75 | 63.19 | 83.87 | 60.47 | 87.61 | 44.22 | 84.23 | 57.84 |
| | **SPADE** | **84.07** | 57.30 | **84.17** | 56.41 | **85.26** | 52.73 | **88.89** | 40.06 | **85.60** | 51.63 |

Table 7: Adversarial samples detection from AA (Croce & Hein, 2020) on CIFAR-10 and CIFAR-100, with perturbation amplitude ranging from $10^{-4}$ to $10^{-1}$ : comparison of SPADE (ResNet teacher) with baselines MSP, MDS, CIDER and KNN using AUC (higher is better) and FPR95 (lower is better). The best results are highlighted in bold.

| | | $\varepsilon = 0.0001$ | | $\varepsilon = 0.001$ | | $\varepsilon = 0.01$ | | $\varepsilon = 0.1$ | | Average | |
|---|---|---|---|---|---|---|---|---|---|---|---|
| | | AUC ↑ | FPR95 ↓ | AUC ↑ | FPR95 ↓ | AUC ↑ | FPR95 ↓ | AUC ↑ | FPR95 ↓ | AUC ↑ | FPR95 ↓ |
| CIFAR-10 | MSP (1) | 82.35 | 74.86 | 82.35 | 74.76 | 82.72 | 73.60 | 87.47 | 58.56 | 83.72 | 70.45 |
| | MDS (3) | 81.61 | 71.04 | 81.61 | 71.13 | 81.83 | 69.63 | 90.26 | 35.47 | 83.83 | 61.82 |
| | KNN (4) | 85.53 | 59.03 | 85.54 | 58.90 | 85.79 | 57.98 | 91.71 | 31.74 | 87.14 | 51.91 |
| | CIDER (5) | **86.10** | **53.72** | **86.10** | **53.56** | **86.34** | **53.09** | 92.00 | 30.86 | **87.53** | **48.26** |
| | **SPADE** | 85.85 | 58.62 | 85.85 | 58.01 | 86.13 | 57.28 | **92.03** | **30.53** | 87.47 | 51.11 |
| CIFAR-100 | MSP (1) | 84.16 | **50.24** | 84.16 | **50.61** | 84.42 | **51.06** | 84.63 | 46.92 | 84.34 | **49.71** |
| | MDS (3) | 81.32 | 61.72 | 81.17 | 61.71 | 81.57 | 61.71 | 82.69 | 70.02 | 81.69 | 63.79 |
| | KNN (4) | 85.19 | 55.12 | 85.18 | 55.58 | 85.45 | 53.98 | 88.18 | 44.36 | 86.00 | 52.26 |
| | CIDER (5) | 83.69 | 59.49 | 83.68 | 59.44 | 84.09 | 59.38 | 86.55 | 47.33 | 84.50 | 56.41 |
| | **SPADE** | **85.71** | 52.90 | **85.70** | 53.14 | **85.96** | 51.91 | **88.45** | 43.31 | **86.46** | 50.32 |

Table 8: Adaptative attacks PGD-ad on CIFAR-10 with perturbation amplitudes ranging from $10^{-4}$ to $10^{-1}$: comparison of SPADE with distance-based baselines MDS and KNN using AUC (higher is better; best in bold) and FPR95 (lower is better; best in bold).

| | $\varepsilon = 0.0001$ | | $\varepsilon = 0.001$ | | $\varepsilon = 0.01$ | | $\varepsilon = 0.1$ | |
|---|---|---|---|---|---|---|---|---|
| | AUC ↑ | FPR95 ↓ | AUC ↑ | FPR95 ↓ | AUC ↑ | FPR95 ↓ | AUC ↑ | FPR95 ↓ |
| MDS (3) | 81.80 | 66.39 | 81.84 | 65.83 | 82.86 | 64.07 | 89.37 | 64.07 |
| KNN (4) | 85.50 | 55.58 | 85.49 | 55.96 | 86.06 | 57.56 | 91.08 | 34.58 |
| SPADE | **86.04** | **55.97** | **86.02** | 57.57 | **86.70** | **55.68** | 91.64 | 32.82 |

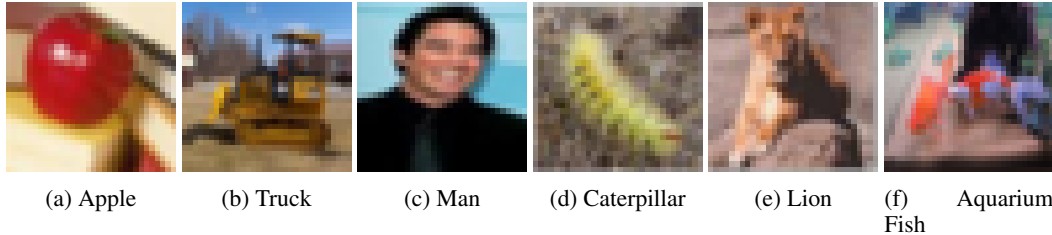

| (a) Apple | (b) Truck | (c) Man | (d) Caterpillar | (e) Lion | (f) Aquarium Fish |

Figure 4: Hard near-OOD setting, with CIFAR-10 as in-distribution and CIFAR-100 as OOD: representative OOD samples missed by SPADE.

We establish that certain guarantees against adversarial OOD attacks can be derived in a manner similar to Theorem 1, contingent on an upper bound on the amplitude of the adversarial attack, as follows.

**Theorem 3.** *Let $G$ be a Generalized Extreme Value (GED) model characterizing the maximum latent distance among an ID sample and its latent nearest ID neighbor:*

$$G(v) = Pr(\|h(\mathbf{x}) - h(\mathbf{x}')\| < v \text{ with } \mathbf{x} \text{ } ID, \mathbf{x}' \text{ nearest neighbor ID of } \mathbf{x})$$

*Similarly, let $G'$ be a GEV model characterizing the minimum latent distance between an OOD sample and its nearest ID:*

$$G'(v') = Pr(\|h(\mathbf{x}) - h(\mathbf{x}')\| > v' \text{ with } \mathbf{x} \text{ } OOD, \mathbf{x}' \text{nearest neighbor ID of } \mathbf{x})$$

*Then let $\mathbf{x}$ be an OOD sample; let $\mathbf{x}'$ be an adversarial perturbation of $\mathbf{x}$ of amplitude $\varepsilon$, such that:*

$$\|\mathbf{x} - \mathbf{x}'\| = \varepsilon$$

*Assume that the input space is Euclidean, and the latent embedding function is Lipschitzian with Lipschitz constant $K$. For*

$$\varepsilon < \frac{1}{K}\left(G'^{-1}(1 - \frac{1}{2}\tau) - G^{-1}(1 - \frac{1}{2}\tau)\right)$$

*$\mathbf{x}'$ is detected with probability $1 - \tau$ as an OOD (i.e. the adversarial OOD attack fails).*

*Proof.* Let $\mathbf{x}$ be an OOD sample, and let $\mathbf{x}'$ denote an adversarial pertubation of $\mathbf{x}$ such that $\|\mathbf{x} - \mathbf{x}'\| = \varepsilon$. Let $\mathbf{x}''$ be the nearest ID neighbor of $\mathbf{x}'$ in the latent space. By contradiction, if $\mathbf{x}'$ is not detected as an OOD sample with probability $1 - \frac{1}{2}\tau$, then:

$$\|h(\mathbf{x}') - h(\mathbf{x}'')\| < G^{-1}(1 - \frac{1}{2}\tau)$$

By the definition of the Lipschitz constant $K$,

$$\|h(\mathbf{x}) - h(\mathbf{x}')\| \leq K\varepsilon$$

Thus, with probability $1 - \frac{1}{2}\tau$,

$$\|h(\mathbf{x}) - h(\mathbf{x}'')\| \leq K\varepsilon + G^{-1}(1 - \frac{1}{2}\tau)$$

By definition of $G'$, with probability $1 - \frac{1}{2}\tau$,

$$G'^{-1}(1 - \frac{1}{2}\tau) \leq \|h(\mathbf{x}) - h(\mathbf{x}'')\|$$

Combining the last two inequalities, it follows that with probability $1 - \tau$,

$$\frac{1}{K}\left(G'^{-1}(1 - \frac{1}{2}\tau) - G^{-1}(1 - \frac{1}{2}\tau)\right) \leq \varepsilon$$

This contradiction concludes the proof. □

Experimentally, the proposed upper bound is non-trivial (i.e., it is positive), as shown on Fig. 5 for in-distribution CIFAR-10. The two adversarial attacks uses SVHN (Fig. 5, left) and MNIST (Fig. 5, right) as OOD datasets. Furthermore, analyzing this bound offers valuable insights into the architectural properties that can enhance the performance of SPADE. For example, Fig. 5 illustrates that the bounds for a ViT model are significantly higher at higher confidence levels compared to a ResNet model (on CIFAR-10).

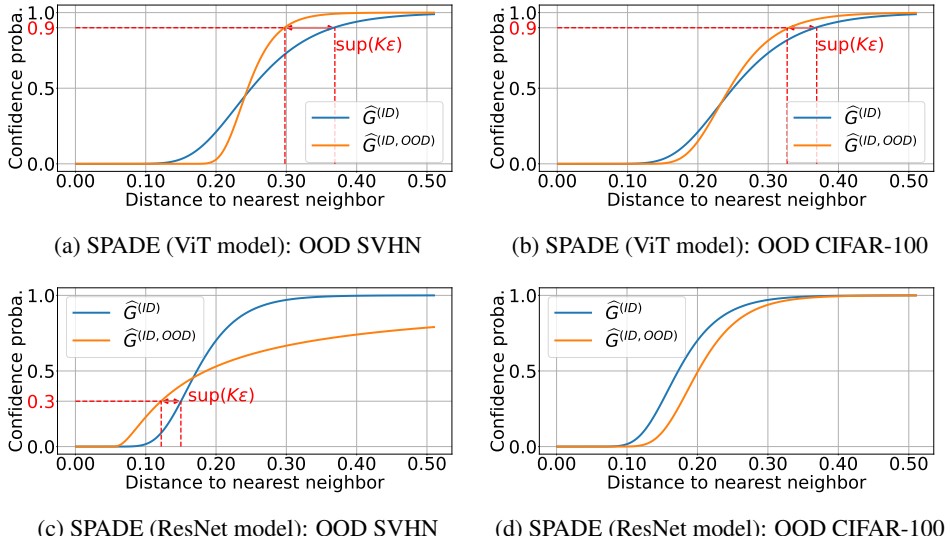

Figure 5: Theorem 3: SPADE Empirical upper bound on $\varepsilon$ for $\tau = 0.1$ for in-distribution CIFAR-10, with teachers ViT (top) and ResNet (bottom). Results are presented for ut-of-distribution datasets SVHN (left) and CIFAR-100 (right). For clarity, we plot $G^{(ID,OOD)} = 1 - G'$.

## C    EXTREME VALUE THEORY

This section provides additional theoretical results related to the Extreme Value Theory, referring the interested reader to Haan & Ferreira (2006) for a comprehensive presentation.

In the following, let $Z$ be a real random variable with cumulative distribution function (CDF) denoted by $F$. We consider $Z^{(n)}$ as the random variable representing the maximum of $n$ independent and identically distributed (i.i.d.) samples of $Z$, i.e., $Z^{(n)} = \max\{Z_1, \ldots, Z_n\}$. The CDF of the maxima, denoted a $F^n$, is given by $Pr(Z^{(n)} < z) = F^n(z)$.

### C.1    GENERALIZED EXTREME VALUE

The key assumption underlying Definition 3 in the main paper is the convergence in distribution of the maxima. To formalize this result, we introduce the following two definitions.

**Definition 6** (Maximum Domain of Attraction). *Let $F$ and $H$ be non-degenerate CDFs. $F$ is said to  be in the domain of attraction of $H$, denoted $F \in \mathcal{DA}(H)$, if there exist sequences of constants $a_n > 0$ and $b_n$ such that every point of continuity $x$ of $H$:*

$$\lim_{n \to \infty} F^n(a_n z + b_n) = H(x) \iff \frac{Z^{(n)} - b_n}{a_n} \to U \qquad (13)$$

*Where $U$ is a random variable whose CDF is $H$.*

**Definition 7** (Distribution Type). *Let $F$ and $G$ be non-degenerate CDFs. $F$ and $G$ are said to be of the same type if there exist constants $a > 0$ and $b$ such that*

$$\forall z \in \mathbb{R}, \quad F(z) = G(az + b) \qquad (14)$$

These definitions play a fundamental role in the Fisher-Tippett theorem, which characterizes the limiting distribution of the maxima of a sequence of independent and identically distributed (i.i.d.) random variables. Informally, this theorem states that if the distribution of the normalized maxima converges, the limit belongs to one of the three classes of distributions. These three classes form the so-called Generalized Extreme Value (GEV) distribution.

**Theorem 4** (Fisher-Tippett). *Let $F$ and $G$ be two non-degenerate CDFs. If $F \in \mathcal{DA}(G)$, then $G$ is of the same type than the following CDF defined for every $z \in \mathbb{R}$*

$$G_{\xi,\mu,\sigma}(z) = \exp \left\{ \begin{array}{ll} \left(1 + \xi\frac{z-\mu}{\sigma}\right)_+^{-1/\xi} & \textit{if } \xi \neq 0 \\ -\exp\left(\frac{\mu-z}{\sigma}\right) & \textit{otherwise} \end{array} \right\} \tag{15}$$

*where $\mu \in \mathbb{R}$ is a location parameter, $\sigma \in \mathbb{R}^+$ is a dispersion parameter, and $\xi \in \mathbb{R}$ is the shape parameter.*

### C.2 GENERALIZED PARETO DISTRIBUTION

As highlighted in the main paper, the Block-Maxima technique becomes inefficient for accurately modeling the tail behavior when the data lacks a natural block structure. An alternative approach is to model tail behavior using threshold exceedances, where the Generalized Pareto Distribution (GPD) plays a central role.

**Definition 8** (Generalised Pareto Distribution). *The* Generalised Pareto Distribution *is defined as:*

$$F_{\xi,\mu,\sigma}(z) = \left\{ \begin{array}{ll} 1 - \exp\left(-\frac{z-\mu}{\sigma}\right) & \textit{if } \xi = 0 \\ 1 - \left(1 + \frac{\xi(z-\mu)}{\sigma}\right)^{-1/\xi} & \textit{otherwise} \end{array} \right. \tag{16}$$

Let us consider the distribution of the excess over a threshold $t \in \mathbb{R}$:

$$F_t(z) = P(Z - t \leq z \mid Z > t) = \frac{F(t+z) - F(t)}{1 - F(t)}, \quad 0 \leq z \leq z_F - t, \tag{17}$$

where $z_F < \infty$ denotes the right endpoint of $F$.

The following theorem establishes that for sufficiently large thresholds $t$, the distribution of excesses over the threshold can be approximated by a Generalized Pareto Distribution (GPD).

**Theorem 5** (Pickands-Balkema-De Haan). *Let $Z$ be a random variable with non-degenerate CDF $F$. There exist positive functions $\alpha(t)$, $\beta(t)$ such that*

$$\lim_{t \to x_F} \left( \sup_{z_F - t} \left| F_t(z) - F_{\xi,\alpha(t),\beta(t)}(z) \right| \right) = 0 \iff F \in \mathcal{DA}(G_\xi) \quad, \quad \xi \in \mathbb{R} \tag{18}$$

*where $\xi$ is the shape parameter, and $G_\xi$ is the Generalized Extreme Value distribution.*

This theorem establishes a formal link between Block Maxima methods and threshold exceedance approaches. Specifically, it provides a theoretical justification for using excess modeling to describe the tail behavior, which is typically captured by Block Maxima methods, that estimate the distribution of extreme values.

## D EXPERIMENTAL SETTING

All experiments are implemented using PyTorch (Paszke et al., 2019) and the OpenOOD package (Zhang et al., 2023)[5].

**Teachers.** All teacher models are trained using stochastic gradient backpropagation, with the learning rate adjusted by the Adam optimizer Kingma & Ba (2015). The models, initially pre-trained on ImageNet, are sourced from the PyTorch hub and fine-tuned (if necessary) on the considered datasets (CIFAR-10, CIFAR-100). This fine-tuning process consists of adding a new fully connected head layer with ReLU activation functions, placed on top of the frozen latent space. The head is trained with a dropout rate of $0.5$.

**Datasets.** OOD samples are categorized into near-OOD and far-OOD distributions. For CIFAR-10, near-OOD samples are from CIFAR-100, while far-OOD samples originate from MNIST, SVHN, Texture, and Places365. For CIFAR-100, near-OOD samples come from ImageNet-1K and Tiny-ImageNet, whereas far-OOD samples are taken from MNIST, SVHN, Texture and Places365. For ImageNet-1K, near-OOD samples are from SSB-Hard (Vaze et al., 2022) and NINCO (Bitterwolf et al., 2023), while far-OOD samples come from iNaturalist (Horn et al., 2018), Texture (Cimpoi et al., 2014), and OpenImage-O (Wang et al., 2022) datasets.

---

[5]https://github.com/Jingkang50/OpenOOD

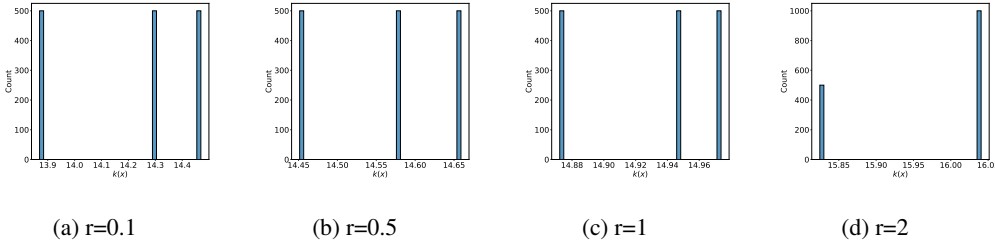

|  (a) r=0.1 | (b) r=0.5 | (c) r=1 | (d) r=2 |

Figure 6: ResNet model: Histogram of local Lipschitz constant $K_r(\boldsymbol{x})$ for different radiuses $r$ on CIFAR-10.

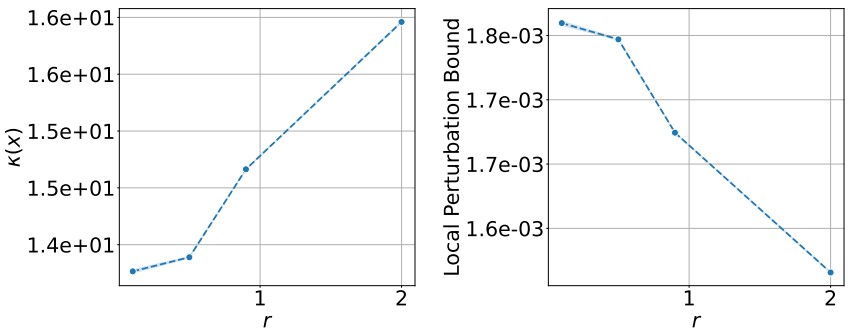

Figure 7: ResNet model, for a representative $\mathbf{x}$: local Lipschitz constant *vs* radius $r$ and upper bound on the adversarial perturbation (Theorem 2).

**Nearest Neighbors.** The nearest neighbor search is performed in the latent space of the model, where the latent representation corresponds to the penultimate layer. Following Sun et al. (2022), a normalization step is applied to the latent features before computing the Euclidean distance, defined as: $(\varphi(\mathbf{x}) = h(\mathbf{x})/\|h(\mathbf{x})\|)$. The distance is then computed as $z = \|\varphi(\mathbf{x}) - \varphi(\mathbf{x}_1)\|$.

Following the approach in Sun et al. (2022), we employ Faiss (Johnson et al., 2019), a library optimized for fast and efficient nearest neighbor search. Specifically, we utilize the `faiss.IndexFlatL2` index, which operates using Euclidean distance. In practice, we precompute the embeddings for all images and store them in a key-value map to facilitate efficient KNN queries.

**Hyperparameters.** The performance comparison with state-of-the-art (SOTA) methods was conducted using the OpenOOD framework (Zhang et al., 2023). Each SOTA method involves one or more hyper-parameters, which are tuned by performing a grid search over the default values provided by OpenOOD. The optimal hyper-parameters are selected by maximizing the area under the curve (AUC) related to the ID-OOD detection, following the common practice (Zhang et al., 2023).

# E  LOCAL PERTURBATION BOUND

The relevance of Theorem 2 (main paper) is illustrated on Figs. 6 and 7 for a ResNet model. Fig. 6 depicts the histogram of the local Lipchitz $K_r$ for $r$ ranging in $\{.1, .5, 1, 2\}$. Fig. 7 shows the local Lipschitz constant $K_r(\mathbf{x})$ for a representative sample $\mathbf{x}$, and the associated lower bound on the adversarial perturbation. Both figures confirm that the lower bound consistently remains smaller than $r$, demonstrating the relevance of Theorem 2.

