# OpenReview forum: "Provably Safeguarding a Classifier from OOD and Adversarial Samples"
_ICLR.cc/2025/Conference — ICLR 2025 Poster_

### Official Review · Reviewer_KzFE · 2024-10-18

**Soundness:** 3
**Presentation:** 3
**Contribution:** 2
**Rating:** 6
**Confidence:** 4

**Summary:**

This paper studies OOD detection. The authors study OOD in terms of GEV, thus enabling a characterization of the probability/confidence of OOD samples. Based on this framework, they propose a method to detect OOD and adversarial examples.

**Strengths:**

The method is well-motivated, and the theory is sound. I love the first-principle design. Paper is well-written and easy to follow. The studied problem is important. Introducing GEV to OOD in this way is novel.

**Weaknesses:**

The main weakness is the weak experimental results. In almost all result tables, the proposed method does not achieve significant benefit over the baselines. In particular, Table 1 shows the proposed method is uniformly dominated by the baseline methods. Table 2 seems to suggest ViT is a particularly strong teacher model for the proposed method, maybe revisiting Table 1 with ViT could bring more benefits.

The authors claim that the proposed methods help to detect adversarial examples. However, the common practice of a defense is to endure adaptive defense, which is aware of the detection and tries to find an adversarial example that surpasses the detection. This highly depends on the claim that the authors would like to make, but overall such experiments would help to establish the significance of the defense.

In summary, while I like the design and the theories, the experimental results do not suggest sufficient improvements, questioning the significance of this work.

**Questions:**

Could the authors provide more convincing experimental results about the benefit of the proposed method?

---

> ### Author Response · Authors · 2024-11-26
>
> Thank you for your encouraging words and for your suggestion:
>
> ## Better Performance w.r.t. OOD detection
> Using the ViT model indeed improves SPADE performance on ImageNet, making it first 6 times out of 10 (and second against MSP otherwise):
>
> |               | SSB ___(AUC)___ | SSB ___(FPR95)___ | NINCO ___(AUC)___ | NINCO ___(FPR95)___ | iNatur. ___(AUC)___ | iNatur. ___(FPR95)___ | Textures ___(AUC)___ | Textures ___(FPR95)___ | OImages ___(AUC)___ | OImages-O ___(FPR95)___ |
> |---------------|------------------|--------------------|---------------|-----------------|---------------------|-----------------------|------------------|--------------------|-----------------------|------------------------|
> | MSP       | **67.31**     |  85.94     |  76.10    |  73.38    |  **85.75**     |   **45.35**     |  82.14     | 56.89   |  **82.28**    | 55.86     |
> | ODIN      |  58.09  |     90.59     |  58.29    |  93.27    |  72.10      |  85.18       |  70.10   |    88.34       |  62.98    |   92.87        |              |
> | KNN       | 36.40            | 96.28            | 36.00         | 94.92           | 34.38               | 91.71                | 48.39        | 86.05         | 42.26            | 89.26             |
> | **SPADE** | 64.55 | **83.52** | **76.91** | **58.58**  | 76.29  | 48.65  | **83.83** | **46.66** | 80.53  | **48.32** |
>
> On all OOD datasets except iNature, the performance is comparable (re AUC) or significantly better (re FPR95) than MSP; SPADE ranks first 2 times out of 4 (and 2nd otherwise) in terms of AUC with a gap of 2.3; it ranks first 4 times out of 4 (and 2nd otherwise) in terms of FPR95, with a gap of 5.93.
> On iNaturalist, our conjecture is that the distinction in latent space w.r.t. ImageNet1k is hardly perceptible, as many classes in iNaturalist actually are subclasses of one class in ImageNet1k (e.g. ImageNet contains one ladybug class, while iNaturalist contains several classes of ladybugs).
> The higher performance of MSP in this setting is attributed to the action of the logit layer.

---

> > ### Author Response · Authors · 2024-11-26
> >
> > ## Better performance w.r.t. adversarial examples
> >
> > Complementary experiments considering harder adversarial attacks have been conducted, including AutoAttack (AA) [2] and Projected Gradient Descent (PGD) [1].
> >
> > These results underline the merit of the first principles grounded behind SPADE's design:
> >
> >
> > W.r.t. PGD attacks [1], SPADE ranks first 5 times out of 6 (and second otherwise) on CIFAR10, and first 4 times out of 6 (and second otherwise) on CIFAR100.
> > W.r.t. AA attacks [2], SPADE ranks first 2 times out of 6 (and second otherwise, behind CIDER) on CIFAR10, and first 4 times out of 6 (and second otherwise behind MSP) on CIFAR100. It is noted that CIDER (best on CIFAR10) ranks 4th 5 times out of 6 on CIFAR100.
> >
> >
> > #### PGD [1] attacks: SPADE performance on CIFAR-10/100 w.r.t SOTA baselines
> >
> >
> > | Dataset | Method    | ε=0.001 ___(AUC)___ | ε=0.001 ___(FPR95)___ | ε=0.01 ___(AUC)___ | ε=0.01 ___(FPR95)___ | ε=0.1 ___(AUC)___ | ε=0.1 ___(FPR95)___ |
> > | ------- | --------- | --------------- | ----------------- | -------------- | ---------------- | ------------- | --------------- |
> > | CIFAR10 | MSP       | 81.69           | 79.10             | 83.19          | 73.58            | 88.82         | 46.44           |
> > |         | MDS       | 81.47           | 69.63             | 82.96          | 62.19            | 90.36         | 35.34           |
> > |         | KNN       | 85.52           | 55.89             | 86.75          | 52.24            | 92.35         | 28.52           |
> > |         | CIDER     | 85.47           | 55.51             | 86.48          | **49.13**        | 91.53         | 31.36           |
> > |         | **SPADE** | **85.66**       | **54.50**         | **86.91**      | 50.89            | **92.49**     | **27.94**       |
> > | |
> > | CIFAR100 | MSP       | 83.25           | **51.38**         | 84.36          | **48.98**        | 86.43         | 42.72           |
> > |         | MDS       | 82.82           | 60.34             | 82.77          | 59.95            | 78.51         | 64.49           |
> > |         | KNN       | 84.04           | 57.78             | 85.12          | 53.14            | 88.77         | 40.73           |
> > |         | CIDER     | 82.75           | 63.19             | 83.87          | 60.47            | 87.61         | 44.22           |
> > |         | **SPADE** | **84.17**       | 56.41             | **85.26**      | 52.73            | **88.89**     | **40.06**       |
> >
> >
> >
> > #### AA [2] attacks: SPADE performance on CIFAR-10/100 w.r.t SOTA baselines
> >
> >
> > | Dataset   | Method   | ε=0.001 ___(AUC)___ | ε=0.001 ___(FPR95)___ | ε=0.01 ___(AUC)___ | ε=0.01 ___(FPR95)___ | ε=0.1 ___(AUC)___ | ε=0.1 ___(FPR95)___ |
> > |-----------|----------|-----------------|-------------------|----------------|------------------|---------------|-----------------|
> > | CIFAR10   | MSP      | 82.35          | 74.76            | 82.72         | 73.60           | 87.47        | 58.56          |
> > |           | MDS      | 81.61          | 71.13            | 81.83         | 69.63           | 90.26        | 35.47          |
> > |           | KNN      | 85.54          | 58.90            | 85.79         | 57.98           | 91.71        | 31.74          |
> > |           | CIDER    | **86.10**      | **53.56**        | **86.34**     | **53.09**       | 92.00        | 30.86          |
> > |           | **SPADE**| 85.85          | 58.01            | 86.13         | 57.28           | **92.03**    | **30.53**      |
> > ||
> > | CIFAR100   | MSP      | 84.16          | **50.61**        | 84.42         | **51.06**       | 84.63        | 46.92          |
> > |           | MDS      | 81.17          | 61.71            | 81.57         | 61.71           | 82.69        | 70.02          |
> > |           | KNN      | 85.18          | 55.58            | 85.45         | 53.98           | 88.18        | 44.36          |
> > |           | CIDER    | 83.68          | 59.44            | 84.09         | 59.38           | 86.55        | 47.33          |
> > |           | **SPADE**| **85.70**      | 53.14            | **85.96**     | 51.91           | **88.45**    | **43.31**      |
> >
> > ---
> >
> > References
> > ==
> > [1] Towards deep learning models resistant to adversarial attacks, Madry et al., ICLR 2018 (https://arxiv.org/abs/1706.06083)
> >
> > [2] Reliable evaluation of adversarial robustness with an ensemble of diverse parameter-free attacks, Croce and Hein, ICML 2020 (https://arxiv.org/abs/2003.01690)

---

> > > ### Author Response · Authors · 2024-11-26
> > >
> > > ## Defense against adaptive attacks
> > >
> > > Trying to assess SPADE vulnerability w.r.t. adversarial OOD detection, we realized that (along the same lines as in Thm 1) a probabilistic guarantee on the detection of adversarial OOD samples, depending on the perturbation amplitude can be established, as follows.
> > >
> > > ### **Thm 2**
> > >
> > > Let $G$ be an generalized extreme value model characterizing the maximum distance among an ID sample and its latent nearest neighbor:
> > >
> > > $G(v) = Pr (\| h(\mathbf{x}) - h(\mathbf{x'})\| < v \text{  with   }  \mathbf{x} \text{ ID}, \mathbf{x'} \text{ ID}, \mathbf{x'} \text{ nearest neighbor of
> > >  } \mathbf{x})$
> > >
> > > Let $G'$ be a GEV model characterizing the minimum latent distance between an OOD sample and its nearest ID:
> > >
> > > $G'(v') = Pr (\| h(\mathbf{x}) - h(\mathbf{x'})\| > v' ~ \text{with}~ \mathbf{x}~OOD, \mathbf{x'} \text{nearest neighbor ID of } \mathbf{x} )$
> > >
> > > Then let $\mathbf{x}$ be an OOD sample; let  $\mathbf{x'}$ be an adversarial perturbation of $\mathbf{x}$ of amplitude $\varepsilon$
> > >
> > > $\| \mathbf{x} - \mathbf{x'} \| = \varepsilon$
> > >
> > > Let us assume that the input space is Euclidean, and the latent embedding is Lipschitzian with Lipschitz constant $K$.
> > > For
> > >
> > > $\varepsilon < \frac{1}{K} \left ( G'^{-1}(1  - \frac{1}{2}\tau) - G^{-1}(1  - \frac{1}{2}\tau) \right )$
> > >
> > > $\mathbf{x'}$ is detected with probability $1 - \tau$ as an OOD (i.e. the adversarial OOD attack fails).
> > >
> > > Proof by contradiction:
> > > Let $\mathbf{x}$ be an OOD sample, and let $\mathbf{x'}$ denote an adversarial pertubation of $\mathbf{x}$ with $\| \mathbf{x} - \mathbf{x'}\| = \varepsilon$.
> > > Let $\mathbf{x''}$ be the (ID) latent nearest neighbor of $\mathbf{x'}$.
> > > By contradiction, if  $\mathbf{x'}$ is not detected as OOD sample with proba $1 - \frac{1}{2}\tau$, then
> > >
> > > $\|h(\mathbf{x'}) - h(\mathbf{x''})\| < G^{-1}(1 -\frac{1}{2}\tau)$
> > >
> > > By definition of $K$,
> > >
> > > $\|h(\mathbf{x}) - h(\mathbf{x'})|\ \leq K \varepsilon$
> > >
> > > Hence with probability $1 - \frac{1}{2}\tau$
> > >
> > > $\|h(\mathbf{x}) - h(\mathbf{x''})\| \leq K\varepsilon +  G^{-1}(1 -\frac{1}{2}\tau)$
> > > By definition of $G'$, with probability $1 - \frac{1}{2}\tau$,
> > >
> > > $G'^{-1}(1  - \frac{1}{2}\tau) \leq \|h(\mathbf{x}) - h(\mathbf{x''})\|$
> > >
> > > From the two previous inequations, it comes: with probability $1 - \tau$,
> > >
> > > $\frac{1}{K} \left ( G'^{-1}(1  - \frac{1}{2}\tau) - G^{-1}(1  - \frac{1}{2}\tau) \right ) \leq \varepsilon$
> > >
> > > This contradiction concludes the proof.
> > >
> > > It remains to show whether this upper bound is not trivial (i.e. is positive). The curve will be added in supplementary material.

---

> > > > ### Author Response · Authors · 2024-11-28
> > > >
> > > > ## Defense against adaptive attacks
> > > >
> > > >
> > > > Following your suggestion, we designed an adaptive attack, called PGD-ad in the following, by augmenting the PGD loss with the distance between $\mathbf{x'}$ and its nearest neighbor (to be minimized). This adaptive attack is assessed on CIFAR-10 (resnet teacher) against SPADE, KNN and MDS, that all rely on latent distances.
> > > >
> > > > For a perturbation amplitude $10^{-2}$, MDS suffers an AUC loss of .1 (from 82.96 to 82.86); KNN suffers an AUC loss of .7 (from 86.75 to 86.06) and SPADE suffers an AUC loss of .2 (from 86.91 to 86.70).
> > > > For a perturbation amplitude $10^{-1}$, MDS suffers an AUC loss of 1; KNN suffers an AUC loss of 1.2  and SPADE suffers an AUC loss of .8 (still ranking first).
> > > >
> > > > These results show the comparative robustness of SPADE:
> > > >
> > > > | Method | ε=0.0001 ___(AUC)___ | ε=0.0001 ___(FPR95)___ | ε=0.001 ___(AUC)___ | ε=0.001 ___(FPR95)___ | ε=0.01 ___(AUC)___ | ε=0.01 ___(FPR95)___ | ε=0.1 ___(AUC)___ | ε=0.1 ___(FPR95)___ |
> > > > |----------|-----------------|-------------------|-----------------|-------------------|----------------|------------------|---------------|-----------------|
> > > > | MDS      | 81.80          | 66.39     | 81.84         | 65.83           | 82.86         | 64.07           | 89.37        | 64.07         |
> > > > | KNN      | 85.50          | **55.58**     | 85.49          | **55.96**            | 86.06         | 57.56           | 91.08        | 34.58          |
> > > > | **SPADE**| **86.04**          | 55.97      | **86.02**          | 57.57           | **86.70**         | **55.68**         | **91.64**    | **32.82**      |
> > > >
> > > >
> > > > For a perturbation amplitude $10^{-3}$, the interaction among the two terms of PGD-ad is much less clear to interpret.
> > > >
> > > >
> > > > Thank you for the suggestion: we hope that this new experiment will confirm the significance of the SPADE defense.

---

> > ### Comment · Reviewer_KzFE · 2024-11-26
> >
> > Thanks for the reply. I found this reply satisfying and would like to ask the authors to incorporate the new results. I decide to raise my score to 6. Reasons for not higher scores: I still found the improvement not uniform, but this is satisfactory enough to me.

---

### Official Review · Reviewer_QvKb · 2024-10-29

**Soundness:** 2
**Presentation:** 3
**Contribution:** 2
**Rating:** 5
**Confidence:** 4

**Summary:**

In this paper, the authors propose sample-efficient probAbilistic detection using extreme value theory (SPADE), which models the training distribution using Extreme Value Theory to create a statistically efficient test that identifies and rejects both OOD and adversarial samples with high probability. Their contributions include a formal OOD definition in relation to a model's latent space, a frugal OOD detection test grounded in EVT, and demonstrated effectiveness against strong baselines across diverse model architectures.

**Strengths:**

1. The paper introduces SPADE, a novel approach leveraging Extreme Value Theory (EVT) for OOD detection, which provides a statistically grounded method to detect and reject OOD samples effectively.

2. SPADE not only detects OOD samples but also offers provable guarantees for rejecting adversarial examples, making it robust against potential adversarial attacks.

3. The approach is experimentally validated across multiple model architectures, demonstrating its versatility and effectiveness compared to strong baselines, enhancing its relevance for various real-world applications.

**Weaknesses:**

1. The provable guarantees only hold on **strong** assumptions. In theorem 1, the authors assume the embedding network is $K$-Lipschitz. Firstly, a $K$-Lipschitz network can already provide provable robustness on adversarial examples. Secondly, due to the complexity of the neural network, it is not possible to calculate the exact Lipschitzness empirically. Thus, this theorem offers limited utility for practical OOD detection scenarios.

2. The experimental setup largely follows standard OOD detection settings and does not address the "provable protection of OOD examples," which is claimed as the main contribution of this work. This raises doubts for the reviewer about whether SPADE can indeed provide provable protection for OOD examples, given the limitations outlined in Weakness 1.

3. SPADE performs poorly on near OOD examples, with Table 1 showing that it is outperformed by the MSP baseline in 3 out of 5 cases. Additionally, the average rank is not an ideal metric. To demonstrate the superiority of the proposed method, SPADE should outperform the baselines across a majority of the datasets.

**Questions:**

See weaknesses

---

> ### Author Response · Authors · 2024-11-22
>
> Thanking you for your insights into the weaknesses of our work, we tried to provide new theoretical and empirical results to overcome these weaknesses:
>
> ### **1.a Relying on Lipschitz is not new**
> You are right, the use of Lipschitz constants w.r.t. adversarial examples is not new:
> In particular, [1] (suggested by Rev EMSd) also relies on the Lipschitz constant of the network, estimated through an extreme value theory approach, to prove robustness.
> In the revised version, we shall cite and discuss [1], as the related work closest  to ours. The comparative novelty and practical utility of SPADE is argued as follows:
>
> * In [1], for each sample $\mathbf{x}$ and each $f_i$ (logit of the classifier), an Extreme Value Theory model is learned to estimate the Lipschitz constant of $f_i$ in $\mathbf{x}$;
> * In SPADE, two EVTs are learned to estimate the minimum latent distance in a class and the maximum latent distance among two distinct classes. Further, this estimation can be made frugal.
>
>
> ### **1.b It is not possible to calculate the exact Lipschitzness empirically; this is a too strong assumption**
> We respectfully disagree: The empirical estimation of the Lipschitz constant of a neural network is investigated for instance in [2,3,4].
>
>
> ### **2. The setup (same as standard OOD detection settings)  does not address the "provable protection of OOD examples"**
> You are right, our experimental setting follows the usual practice (wouldn't it be a weakness to do otherwise ?)
> However, your critique about the provable detection of OOD samples prompted us to  establish a new formal result, providing a proof of protection against OOD examples.
> Formally, we establish that the adversarial perturbation of an OOD sample is detected with probability $1 - \tau$ if the perturbation amplitude $\varepsilon$ is less than the following upper bound:
>
> $\varepsilon < \frac{1}{K} \left ( G'^{-1}(1  - \frac{1}{2}\tau) - G^{-1}(1  - \frac{1}{2}\tau) \right )$
>
> where $G$ denotes the generalized extreme value model characterizing the maximum distance among an ID sample and its latent nearest neighbor:
>
> $G(v) = Pr (\| h(\mathbf{x}) - h(\mathbf{x'})\| < v \text{ }|\text{ }  \mathbf{x} \text{ ID}, \mathbf{x'} \text{ ID}, \mathbf{x'} \text{ nearest neighbor of } \mathbf{x})$
>
> and $G'$ denotes the GEV model characterizing the minimum distance between an ID and an OOD sample:
>
> $G'(v') = Pr (\| h(\mathbf{x}) - h(\mathbf{x'})\| > v' \text{ }|\text{ }  \mathbf{x} \text{ ID}, \mathbf{x'} \text{ OOD} )$
>
> The utility of this result (the fact that the upper bound is positive) will be experimentally assessed in supplementary material.
>
> ### **3. SPADE performs poorly on near OOD examples (Table 1)**
>
> Acknowledging your criticism, we considered more challenging near-OOD examples, based on [5] and following Rev 3aNG:
>
>  We understand the importance of considering a more challenging near-OOD task, using CIFAR10 as in-distribution vs CIFAR10 as Out-Of-Distribution, and vice versa. We followed the same experimental setting  in [5], Table 2: we used a ViT-B_16 teacher model trained on ImageNet-1k with fine-tuning on the in-distribution. The results are displayed below (more in SM). For CIFAR10 in-distribution, SPADE ranks first in AUC (second for FPR95); for CIFAR100 in-distribution, SPADE ranks 2nd in AUC (third in FPR95).
>
>
> | In-Distribution | CIFAR10 ___(AUC)___ | CIFAR10 ___(FPR95)___ | CIFAR100 ___(AUC)___ | CIFAR100 (FPR95) |
> |------------------|----------------|-------------------|------------------|-------------------|
> | MSP             | 92.01          | **27.66**        | 76.50           | 70.68             |
> | MDS             | 87.08          | 53.37            | **78.02**       | **66.28**         |
> | KNN             | 91.52          | 35.83            | 69.80           | 85.18             |
> | CIDER           | 60.34          | 90.03            | 60.09           | 89.87             |
> | **SPADE**       | **92.58**      | 32.53            | 77.32           | 71.69             |
>
>
> Some representative failure cases are illustrated in SM (Fig. 2), showing diverse OOD samples (from CIFAR100) mistaken as in-distribution (CIFAR10).
>
> ## References
>
> [1] Evaluating the robustness of neural networks: An extreme value theory approach, Weng et al., ICLR 2018
>
> [2] Fazlyab, M., Robey, A., Hassani, H., Morari, M., & Pappas, G. J. (2019). Efficient and Accurate Estimation of Lipschitz Constants for Deep Neural Networks. NeurIPS
>
> [3] M. Serrurier, F. Mamalet, A. González-Sanz, T. Boissin, J. -M. Loubes and E. del Barrio (2021). Achieving robustness in classification using optimal transport with hinge regularization . CVPR
>
> [4] Latorre, F., Rolland, P., & Cevher, V. (2020). Lipschitz constant estimation of Neural Networks via sparse polynomial optimization.ICLR
>
> [5] Adversarial vulnerability of powerful near out-of-distribution detection. Stanislav Fort (https://arxiv.org/abs/2201.07012)

---

> ### Comment · Reviewer_QvKb · 2024-11-26
> **Official Comment by Reviewer QvKb**
>
> Thanks for the rebuttal. Since my concerns about the lipschitzness and the provable protection of OOD examples are not addressed, I will keep my rating.
>
> 1 b). [2-4] can only be used to estimate the Lipschitzness of small scaled neural networks. They cannot be adapted to the large-scale NN, e.g., transformers and LLM due to the computational cost. Thus, the assumption is too strong.
>
> 2. As the provable guarantee rely on the Lipschitzness, which is not accessible for DNN, the practicality of the proposed bound remains unclear.

---

> > ### Author Response · Authors · 2024-11-28
> >
> > May we respectfully disagree ?
> > After [1], the CLEVER method, which uses Generalized Extreme Value models to estimate Lipschitz constants in every training sample, "is computationally feasible even for large networks like Inception-v3".
> >
> > For context, Inception-v3 comprises circa $24×10^6$ parameters, underscoring CLEVER's scalability.
> >
> >
> >
> > [1] Evaluating the robustness of neural networks: An extreme value theory approach, Weng et al., ICLR 2018

---

### Official Review · Reviewer_EMSd · 2024-10-30

**Soundness:** 3
**Presentation:** 3
**Contribution:** 2
**Rating:** 6
**Confidence:** 4

**Summary:**

This paper uses extreme value theory as a means to detect out-of-distribution (OOD) samples. By abstaining to predict on samples that are determined to be OOD, the proposed approach is shown to yield high-probability robustness guarantees against adversarial attacks up to a certain perturbation magnitude. Experiments are conducted on CIFAR-10, CIFAR-100, and ImageNet to assess the performance of the proposed method compared to prior OOD detection schemes.

**Strengths:**

The problem considered is interesting, and fits well within the scope of the ICLR community. The paper is easy to follow, and does a good job at clearly yet concisely introducing the main tools being used (e.g., extreme value theory). There is a nice blend of conceptual/theoretical development, and empirical evaluation.

**Weaknesses:**

See my specific Questions below.

**Questions:**

1. Line 90: You use the acronym ID before defining it.
2. Line 102: You should de-italicize the term $\eta$-invariant that you are definining.
3. Definition 3: The role of the parameters $\mu,\sigma,\zeta$ are unclear in how you wrote Definition 3. It seems to me like you need to re-word things to somehow say that "there exist $\mu,\sigma,\zeta$ such that $P(Z^{(\ell)} < z) \to G_{\zeta,\mu,\sigma}(z)$." Otherwise, the reader might be inclined to think that the choice of parameters is somehow "up to us."
4. Line 150: Extreme value theory has been utilized by the ML community in popular robustness verification works such as [1]. I encourage you to take a look at [1] as well as the references therein.
5. Line 169: "For $Y = c$, let $Z_c$ be the random variable defined as the distance between $h(X)$ and its $k$-th nearest neighbor in latent distance, belonging to $\mathcal{D}$ with same class $c$." It would help readability if you write out the mathematical expression defining the random variable $Z_c$.
6. Line 170: What is $k$? Is this a user-specified hyperparameter of your method?
7. Line 171: The notation here is inconsistent; you should be writing $Z^{(\ell)}$, not $Z^{(l)}$. Also, before, you used $P$ for probability, not $Pr$.
8. Line 215: Where is the minimum coming into this equation defining the extreme value distribution $G^{(c,c')}$? It seems like without the minimum inside the probability expression on the right, you aren't really considering extreme values.
9. Theorem 1: It looks like now you are assuming that the instance space $\mathcal{X}$ is a Euclidean space $\mathbb{R}^d$ (so that you can define norms on the instance space as well as Lipschitzness of the embedding map), which was not explicitly assumed before. Therefore, you should make this clear in the theorem statement.
10. Theorem 1: Do you mean to say $f_\tau$ instead of $f_\epsilon$ in the statement of the theorem?
11. Experimental Setting: You state that you generate attacks using FGSM. At this point FGSM, as introduced in Goodfellow et al., 2015, is not really considered by the adversarial robustness community to be a strong attack. At the very least, I'd expect you to use projected gradient descent (PGD) [2], or something stronger like AutoAttack [3] which is used in the RobustBench benchmarks. Have you tested your methods against these stronger attacks?
12. There are a handful of distinct conceptual discussions and innovations leading up to your experiments in Sections 4 and 5. However, there is no explicit description of SPADE where you bring everything together into a single model/algorithm. I think it would greatly benefit readability if you were to conclude your conceptual discussions by writing out SPADE as an explicit algorithm (including any robustness certification steps), immediately before moving into experiments.
13. Line 318: I'm assuming by "KNN" you are referring to $k$-nearest neighbors? You should clearly define this acronym before using it. Similarly, MSP is not defined. Please define it, or at the very least, associate a reference to it.
14. Line 190 and Experiments: On line 190 you said that you would tease out the effect of the latent embeddings on the OOD tests in Section 5 (since your proposed OOD test in Definition 4 depends on the embedding from instance space to latent space). However, after reading through the experiments of Section 5, I do not see how you are answering this concern. Could you clarify?

[1] "Evaluating the robustness of neural networks: An extreme value theory approach," Weng et al., ICLR 2018

[2] "Towards deep learning models resistant to adversarial attacks," Madry et al., ICLR 2018

[3] "Reliable evaluation of adversarial robustness with an ensemble of diverse parameter-free attacks," Croce and Hein, ICML 2020

---

> ### Author Response · Authors · 2024-11-22
>
> We thank you for your detailed comments and questions, that will significantly help us to improve the paper and its readability.
>
> ---
>
> **1-2-3-5-7-13:** Thank you: indeed we shall correct the typos and take these criticisms and comments in the revised version.
>
> ---
>
> **4**: Thank you for pointing at reference [1], that will be cited and discussed. We understand the difference as follows: CLEVER proposes (Thm 3.2) a lower bound for an untargeted attack in a point $x_0$, depending on the Lipschitz constants of the classifier heads in $x_0$.
> The EVT is leveraged (Thm 4.1) to estimate these Lipschitz constants in each $x_0$.
> SPADE also proposes an attack-agnostic score. The difference is that the EVT models used in SPADE relate to the maximum latent distance between two in-distribution samples in the same class (respectively the minimum latent distance between two in-istribution samples in different classes); the overall number of EVT models thus is constant in SPADE, as opposed to linear in the number of samples in CLEVER. Further, the frugality of estimation of the SPADE EVT models is demonstrated in the experiments.
>
> ---
>
> **6**: $k$ is a hyper-parameter of the approach (as in $k$-nearest neighbors); $k$ is set to 1 in the experiments. The analysis shows a low sensitivity w.r.t. $k$.
>
> ---
>
> **9** : You are right. The assumption that the input space is a Euclidean space will be added to Thm 1.
>
> ---
>
> **10**: You are right, $f_\epsilon$ should be $f_\tau$
>
> ---
>
>
> **12**: We added the pseudo-code of the algorithm at the end of section 3, in the revised paper.
>
> ---
>
> **14**: The sensitivity w.r.t. the teacher actually is the question Q2 (goals of experiments, l. 289). The experimental answer (l. 356 and seq.) is that the results are generally much better with a transformer teacher architecture. Our tentative interpretation, based on the comparison with CIDER, is actually formulated in footnote 3: what matters is i) the variation and informativeness of the teacher (diameter of a class manifold, distance among two class manifolds); ii) the Lipschitz constant of the latent embedding. Further work will be concerned with testing this conjecture, by considering a new loss, augmenting the CIDER loss with a term restricting the Lipschitz constant of the latent embedding.

---

> > ### Author Response · Authors · 2024-11-22
> >
> > **11**: Complementary experiments against stronger adversarial attacks (PGD and AutoAttack) have been conducted:
> >
> > #### PGD [2] attacks: SPADE performance on CIFAR-10/100 w.r.t SOTA baselines
> >
> > | Dataset | Method    | ε=0.001 ___(AUC)___ | ε=0.001 ___(FPR95)___ | ε=0.01 ___(AUC)___ | ε=0.01 ___(FPR95)___ | ε=0.1 ___(AUC)___ | ε=0.1 ___(FPR95)___ |
> > | ------- | --------- | --------------- | ----------------- | -------------- | ---------------- | ------------- | --------------- |
> > | CIFAR10 | MSP       | 81.69           | 79.10             | 83.19          | 73.58            | 88.82         | 46.44           |
> > |         | MDS       | 81.47           | 69.63             | 82.96          | 62.19            | 90.36         | 35.34           |
> > |         | KNN       | 85.52           | 55.89             | 86.75          | 52.24            | 92.35         | 28.52           |
> > |         | CIDER     | 85.47           | 55.51             | 86.48          | **49.13**        | 91.53         | 31.36           |
> > |         | **SPADE** | **85.66**       | **54.50**         | **86.91**      | 50.89            | **92.49**     | **27.94**       |
> > | |
> > | CIFAR100 | MSP       | 83.25           | **51.38**         | 84.36          | **48.98**        | 86.43         | 42.72           |
> > |         | MDS       | 82.82           | 60.34             | 82.77          | 59.95            | 78.51         | 64.49           |
> > |         | KNN       | 84.04           | 57.78             | 85.12          | 53.14            | 88.77         | 40.73           |
> > |         | CIDER     | 82.75           | 63.19             | 83.87          | 60.47            | 87.61         | 44.22           |
> > |         | **SPADE** | **84.17**       | 56.41             | **85.26**      | 52.73            | **88.89**     | **40.06**       |
> >
> >
> >
> > #### AA [3] attacks: SPADE performance on CIFAR-10/100 w.r.t SOTA baselines
> >
> >
> > | Dataset   | Method   | ε=0.001 ___(AUC)___ | ε=0.001 ___(FPR95)___ | ε=0.01 ___(AUC)___ | ε=0.01 ___(FPR95)___ | ε=0.1 ___(AUC)___ | ε=0.1 ___(FPR95)___ |
> > |-----------|----------|-----------------|-------------------|----------------|------------------|---------------|-----------------|
> > | CIFAR10   | MSP      | 82.35          | 74.76            | 82.72         | 73.60           | 87.47        | 58.56          |
> > |           | MDS      | 81.61          | 71.13            | 81.83         | 69.63           | 90.26        | 35.47          |
> > |           | KNN      | 85.54          | 58.90            | 85.79         | 57.98           | 91.71        | 31.74          |
> > |           | CIDER    | **86.10**      | **53.56**        | **86.34**     | **53.09**       | 92.00        | 30.86          |
> > |           | **SPADE**| 85.85          | 58.01            | 86.13         | 57.28           | **92.03**    | **30.53**      |
> > ||
> > | CIFAR100   | MSP      | 84.16          | **50.61**        | 84.42         | **51.06**       | 84.63        | 46.92          |
> > |           | MDS      | 81.17          | 61.71            | 81.57         | 61.71           | 82.69        | 70.02          |
> > |           | KNN      | 85.18          | 55.58            | 85.45         | 53.98           | 88.18        | 44.36          |
> > |           | CIDER    | 83.68          | 59.44            | 84.09         | 59.38           | 86.55        | 47.33          |
> > |           | **SPADE**| **85.70**      | 53.14            | **85.96**     | 51.91           | **88.45**    | **43.31**      |
> >
> >
> > ---
> >
> > We hope that our answers address your concerns and we thank you.
> >
> > ## References
> >
> > [2] Towards deep learning models resistant to adversarial attacks, Madry et al., ICLR 2018 (https://arxiv.org/abs/1706.06083)
> >
> > [3] Reliable evaluation of adversarial robustness with an ensemble of diverse parameter-free attacks, Croce and Hein, ICML 2020 (https://arxiv.org/abs/2003.01690)

---

> > > ### Comment · Reviewer_EMSd · 2024-11-22
> > >
> > > I thank the authors for their thorough responses and new experiments. I have decided to increase my overall Rating by 1 point.

---

### Official Review · Reviewer_3aNg · 2024-11-12

**Soundness:** 2
**Presentation:** 2
**Contribution:** 3
**Rating:** 5
**Confidence:** 3

**Summary:**

The paper presents SPADE (Sample-efficient Probabilistic Detection using Extreme Value Theory) = a new method for detecting out-of-distribution (OOD) and adversarial samples in neural network classifiers. The key contributions are:
a. A novel method to transform a classifier into an "abstaining classifier" that can refuse to make predictions on suspicious inputs
b. Use of Extreme Value Theory (EVT) to model the training distribution in the classifier's latent space
c. Mathematical guarantees for detecting both OOD and adversarial samples
d. Experimental validation across multiple architectures (ResNet, VGG, ViT) and datasets

I am pretty unfamiliar with Extreme Value Theory, so please flag if if I'm misunderstanding something basic in my review. But I'm very familiar with OOD detection and adversarial attacks, so it could compensate for it.

**Strengths:**

1. Theoretical Foundation: The approach is grounded in statistical theory (EVT)
2. Sample Efficiency: Good performance even with strong subsampling of the train set
3. Versatility: Works with different architectures and can handle both OOD and adversarial inputs
4. Evaluation: Tested against strong baselines on multiple datasets

**Weaknesses:**

# 1. Missing challenging near-OOD evaluation
The evaluations in the Table 2 and Table 3 are missing a very simple, yet challenging near OOD detection task which is CIFAR-100 vs CIFAR-10 (and the other way round). [1] shows strong results for large models and also provides a human score to benchmark against. I would be very curious to see the AUROC, AUC and FPR95 on that. I believe that this would be a very worthwhile and easy evaluation to add and I think the authors should try it. The AUROC in particular is something I would really like to see. Also a set of examples of where the model fails. **I will consider increasing my score if the authors successfully address this point.**

# 2. A collection of weakness that do not need to be addressed in the rebuttal:
2.1 Computational Complexity: The GEV model is quadratic in the number of samples
2.2 Performance Trade-off: While SPADE performs well in general, it's sometimes outperformed by simpler methods like KNN for specific tasks
2.3. Parameter Sensitivity: The effectiveness depends on the choice of teacher model and its latent space characteristics
2.4. High FPR95 Values: Shows tendency to be overly cautious. Perhaps rejecting valid samples?

# 3. Weak adversarial attacks used only
The paper's evaluation of adversarial robustness is limited. They only test against the FGSM attack (=very basic) with small perturbation magnitudes (epsilon from 0.001 to 0.004). Given that SPADE is positioned as providing security guarantees against adversarial examples, a more thorough evaluation against SOTA attacks would be necessary to support the claims. Additionally, the authors should clarify whether their evaluation considers attacks targeting just the classifier or the full system including the OOD detector, and include both targeted and untargeted attack scenarios. **This is especially relevant to the next weak point.**

# 4. White-box attacks

I have the following concern that I would like you to address directly:

1. SPADE offers an elegant theoretical framework for OOD detection based on a distance metric in the latent space
2. [1] shows that distance metrics in the latent space can be amazing at even hard, near-OOD detection
3. [2] shows that white-box directly targeting such a metric can *still completely destroy it* with a simple adversarial attack

Given SPADE's reliance on distance metrics in latent space and GEV models, it may be vulnerable to white-box attacks that:

4. Directly optimize perturbations to minimize latent space distances to k-nearest neighbors while maintaining misclassification
5. Exploit knowledge of the GEV model parameters to generate samples that appear in-distribution
6. Target the gap between separate class-specific GEV models to find adversarial blind spots

The authors should consider and discuss these potential vulnerabilities, particularly since they claim SPADE as providing security guarantees. While Theorem 1 provides a lower bound on adversarial perturbation magnitude, an analysis of potential attack strategies and additional defensive measures (such as ensemble approaches or adversarial training) would strengthen the paper's security claims.

# References

[1] Exploring the Limits of Out-of-Distribution Detection. Stanislav Fort, Jie Ren, Balaji Lakshminarayanan (https://arxiv.org/abs/2106.03004)

[2] Adversarial vulnerability of powerful near out-of-distribution detection. Stanislav Fort (https://arxiv.org/abs/2201.07012)

**Questions:**

Included in the weaknesses section.

I am primarily interested in the authors answers to:
a. the CIFAR-100 <-> CIFAR-10 challenging near OOD task(s)
b. the white-box attack on latent metrics point

---

> ### Author Response · Authors · 2024-11-20
>
> Thank you for your inspiring and demanding feedback, allowing us to clarify the scope of our work. Please find below our responses to your first points:
>
>
> ---
>
> ### **1. Missing challenging near-OOD evaluation (CIFAR-100 vs CIFAR-10)**
>
> Thank you for pointing at [1] and [2], that we shall cite and discuss in the revised version. We understand the importance of considering a more challenging near-OOD task, using CIFAR10 as in-distribution vs CIFAR10 as Out-Of-Distribution, and vice versa. We followed the same experimental setting as in [1], Table 2: we used a ViT-B_16 teacher model trained on ImageNet-1k with fine-tuning on the in-distribution. The results are displayed below (more in SM). For CIFAR10 in-distribution, SPADE ranks first in AUC (second for FPR95); for CIFAR100 in-distribution, SPADE ranks 2nd in AUC (third in FPR95).
>
>
> | In-Distribution | CIFAR10 ___(AUC)___ | CIFAR10 ___(FPR95)___ | CIFAR100 ___(AUC)___ | CIFAR100 (FPR95) |
> |------------------|----------------|-------------------|------------------|-------------------|
> | MSP             | 92.01          | **27.66**        | 76.50           | 70.68             |
> | MDS             | 87.08          | 53.37            | **78.02**       | **66.28**         |
> | KNN             | 91.52          | 35.83            | 69.80           | 85.18             |
> | CIDER           | 60.34          | 90.03            | 60.09           | 89.87             |
> | **SPADE**       | **92.58**      | 32.53            | 77.32           | 71.69             |
>
>
> Some representative failure cases are illustrated in SM (Fig. 2), showing diverse OOD samples (from CIFAR100) mistaken as in-distribution (CIFAR10).
>
>
> ### **2. Considering Stronger Adversarial Attacks**
>
> To address your remark, we conducted complementary experiments using PGD [3] and AutoAttack [4] to generate adversarial examples, with perturbation amplitude $\varepsilon$ ranging in $10^{-3}, 10^{-2}, 10^{-1}$ (more in SM, Tables 3 and 4).
> On CIFAR10, SPADE ranks first against the baselines (except for FPR95, $\varepsilon = 10^{-2}$).
> On CIFAR100, SPADE ranks first w.r.t. AUC for all $\varepsilon$ values, and second (or first) in FPR95.
> This confirms SPADE's robustness in front of (more challenging) attacks.
>
> #### PGD [3] attacks (more in Table 3 SM)
>
> | Dataset | Method    | ε=0.001 ___(AUC)___ | ε=0.001 ___(FPR95)___ | ε=0.01 ___(AUC)___ | ε=0.01 ___(FPR95)___ | ε=0.1 ___(AUC)___ | ε=0.1 ___(FPR95)___ |
> | ------- | --------- | --------------- | ----------------- | -------------- | ---------------- | ------------- | --------------- |
> | CIFAR100 | MSP       | 83.25           | **51.38**         | 84.36          | **48.98**        | 86.43         | 42.72           |
> |         | MDS       | 82.82           | 60.34             | 82.77          | 59.95            | 78.51         | 64.49           |
> |         | KNN       | 84.04           | 57.78             | 85.12          | 53.14            | 88.77         | 40.73           |
> |         | CIDER     | 82.75           | 63.19             | 83.87          | 60.47            | 87.61         | 44.22           |
> |         | **SPADE** | **84.17**       | 56.41             | **85.26**      | 52.73            | **88.89**     | **40.06**       |
>
>
>
> #### AA [4]  attacks (more in Table 4 SM)
>
>
> | Dataset   | Method   | ε=0.001 ___(AUC)___ | ε=0.001 ___(FPR95)___ | ε=0.01 ___(AUC)___ | ε=0.01 ___(FPR95)___ | ε=0.1 ___(AUC)___ | ε=0.1 ___(FPR95)___ |
> |-----------|----------|-----------------|-------------------|----------------|------------------|---------------|-----------------|
> | CIFAR100   | MSP      | 84.16          | **50.61**        | 84.42         | **51.06**       | 84.63        | 46.92          |
> |           | MDS      | 81.17          | 61.71            | 81.57         | 61.71           | 82.69        | 70.02          |
> |           | KNN      | 85.18          | 55.58            | 85.45         | 53.98           | 88.18        | 44.36          |
> |           | CIDER    | 83.68          | 59.44            | 84.09         | 59.38           | 86.55        | 47.33          |
> |           | **SPADE**| **85.70**      | 53.14            | **85.96**     | 51.91           | **88.45**    | **43.31**      |
>
> ---
> ### **Conclusion**
>
> We hope that these additional results address some of your criticisms, and we look forward to your feedback.
>
> ---
>
> ### References
>
> [1] Exploring the Limits of Out-of-Distribution Detection. Stanislav Fort, Jie Ren, Balaji Lakshminarayanan, NeurIPS 2021 (https://arxiv.org/abs/2106.03004)
>
> [2] Adversarial vulnerability of powerful near out-of-distribution detection. Stanislav Fort (https://arxiv.org/abs/2201.07012)
>
> [3] Towards deep learning models resistant to adversarial attacks, Madry et al., ICLR 2018 (https://arxiv.org/abs/1706.06083)
>
> [4] Reliable evaluation of adversarial robustness with an ensemble of diverse parameter-free attacks, Croce and Hein, ICML 2020 (https://arxiv.org/abs/2003.01690)

---

> ### Author Response · Authors · 2024-11-22
>
> ### **3 White-box attacks on latent metrics**
>
> Thank you! Trying to assess SPADE vulnerability w.r.t. adversarial OOD detection (as discussed by [2]), we realized that (along the same lines as in Thm 1) a probabilistic guarantee on the detection of adversarial OOD samples, depending on the perturbation amplitude can be established, as follows.
>
>
> ### **Thm 2**
>
> Let $G$ be an generalized extreme value model characterizing the maximum distance among an ID sample and its latent nearest neighbor:
>
> $G(v) = Pr (\| h(\mathbf{x}) - h(\mathbf{x'})\| < v \text{  with   }  \mathbf{x} \text{ ID}, \mathbf{x'} \text{ ID}, \mathbf{x'} \text{ nearest neighbor of
>  } \mathbf{x})$
>
> Let $G'$ be a GEV model characterizing the minimum latent distance between an OOD sample and its nearest ID:
>
> $G'(v') = Pr (\| h(\mathbf{x}) - h(\mathbf{x'})\| > v' ~ \text{with}~ \mathbf{x}~OOD, \mathbf{x'} \text{nearest neighbor ID of } \mathbf{x} )$
>
> Then let $\mathbf{x}$ be an OOD sample; let  $\mathbf{x'}$ be an adversarial perturbation of $\mathbf{x}$ of amplitude $\varepsilon$
>
> $\| \mathbf{x} - \mathbf{x'} \| = \varepsilon$
>
> Let us assume that the input space is Euclidean, and the latent embedding is Lipschitzian with Lipschitz constant $K$.
> For
>
> $\varepsilon < \frac{1}{K} \left ( G'^{-1}(1  - \frac{1}{2}\tau) - G^{-1}(1  - \frac{1}{2}\tau) \right )$
>
> $\mathbf{x'}$ is detected with probability $1 - \tau$ as an OOD (i.e. the adversarial OOD attack fails).
>
> Proof by contradiction:
> Let $\mathbf{x}$ be an OOD sample, and let $\mathbf{x'}$ denote an adversarial pertubation of $\mathbf{x}$ with $\| \mathbf{x} - \mathbf{x'}\| = \varepsilon$.
> Let $\mathbf{x''}$ be the (ID) latent nearest neighbor of $\mathbf{x'}$.
> By contradiction, if  $\mathbf{x'}$ is not detected as OOD sample with proba $1 - \frac{1}{2}\tau$, then
>
> $\|h(\mathbf{x'}) - h(\mathbf{x''})\| < G^{-1}(1 -\frac{1}{2}\tau)$
>
> By definition of $K$,
>
> $\|h(\mathbf{x}) - h(\mathbf{x'})|\ \leq K \varepsilon$
>
> Hence with probability $1 - \frac{1}{2}\tau$
>
> $\|h(\mathbf{x}) - h(\mathbf{x''})\| \leq K\varepsilon +  G^{-1}(1 -\frac{1}{2}\tau)$
> By definition of $G'$, with probability $1 - \frac{1}{2}\tau$,
>
> $G'^{-1}(1  - \frac{1}{2}\tau) \leq \|h(\mathbf{x}) - h(\mathbf{x''})\|$
>
> From the two previous inequations, it comes: with probability $1 - \tau$,
>
> $\frac{1}{K} \left ( G'^{-1}(1  - \frac{1}{2}\tau) - G^{-1}(1  - \frac{1}{2}\tau) \right ) \leq \varepsilon$
>
> This contradiction concludes the proof.
>
> It remains to show whether this upper bound is not trivial (i.e. is positive). The curve will be added in supplementary material.

---

> > ### Author Response · Authors · 2024-11-28
> >
> > ## Defense against adaptive attacks
> >
> >
> > Following your suggestion, we designed an adaptive attack, called PGD-ad in the following, by augmenting the PGD loss with the distance between $\mathbf{x'}$ and its nearest neighbor (to be minimized). This adaptive attack is assessed on CIFAR-10 (resnet teacher) against SPADE, KNN and MDS, that all rely on latent distances.
> >
> > For a perturbation amplitude $10^{-2}$, MDS suffers an AUC loss of .1 (from 82.96 to 82.86); KNN suffers an AUC loss of .7 (from 86.75 to 86.06) and SPADE suffers an AUC loss of .2 (from 86.91 to 86.70).
> > For a perturbation amplitude $10^{-1}$, MDS suffers an AUC loss of 1; KNN suffers an AUC loss of 1.2  and SPADE suffers an AUC loss of .8 (still ranking first).
> >
> > These results show the comparative robustness of SPADE:
> >
> > | Method | ε=0.0001 ___(AUC)___ | ε=0.0001 ___(FPR95)___ | ε=0.001 ___(AUC)___ | ε=0.001 ___(FPR95)___ | ε=0.01 ___(AUC)___ | ε=0.01 ___(FPR95)___ | ε=0.1 ___(AUC)___ | ε=0.1 ___(FPR95)___ |
> > |----------|-----------------|-------------------|-----------------|-------------------|----------------|------------------|---------------|-----------------|
> > | MDS      | 81.80          | 66.39     | 81.84         | 65.83           | 82.86         | 64.07           | 89.37        | 64.07         |
> > | KNN      | 85.50          | **55.58**     | 85.49          | **55.96**            | 86.06         | 57.56           | 91.08        | 34.58          |
> > | **SPADE**| **86.04**          | 55.97      | **86.02**          | 57.57           | **86.70**         | **55.68**         | **91.64**    | **32.82**      |
> >
> >
> > For a perturbation amplitude $10^{-3}$, the interaction among the two terms of PGD-ad is much less clear to interpret.
> >
> >
> > Thank you for the suggestion: we hope that this new experiment will confirm the significance of the SPADE defense.

---

### Meta-Review · Area_Chair_mUBz · 2024-12-21

**Metareview:**

This method develops SPADES, a method that turns a classifier into one with the possibility of abstention in the presence of out-of-distribution (OOD) samples. The approach is based on Extreme Value Theory, and can - as a by-product - function as a defense against adversarial examples. The paper presents results showing that an attack will be detected by their method with high probability, under suitable assumptions. Numerical experiments support their claims.

**Strengths**
* Clear presentation and methodology
* Nice and simple observations
* Application to both OOD and adversarial examples with diverse set of experiments.

**Weaknesses**
* While the experiments are thorough and extensive (particularly after the rebuttal), the authors show that their method is *somewhat* effective against other competitive methods, but the improvement over these others is modest.
* The work stresses the qualifier of "*provable defenses*". However, the results depend on the estimation of the Lipschitz constant of the network. There is no discussion in the paper as to how this quantity can be provably estimated for the cases and architectures considered in this work so that the guarantee that they provide actually applies in practice.

**Summary**

This work has received mixed reviewers, and it is truly a borderline case. The AC spent considerable time reading all the comments, responses, and the paper itself. In all, I believe the authors have done a good job of increasing the quantity and quality of the experimental evidence provided.
I am recommending acceptance *provided* that the authors make this limitation very clear in their revision, explaining the extent to which the Lipschitz constant can be provably estimated (e.g. by means of the related works that they cite). If this is not the case, the qualifier '*provably*' should be removed from the title.

**Additional Comments On Reviewer Discussion:**

All reviewers had insightful comments. Most comments and questions pertained to the (initially) weak comparison with stronger adversarial attacks (e.g. PGD and AutoAttack), as well as the detection of OOD for smaller/closer shifts (e.g. CIFAR10 vs CIFAR100). All of these were successfully addressed, and the authors have done a good job at significantly expanding their experimental results.

Rev QvKb raised the issue of needing to estimate the Lipschitz constant for their method to be a provable (probabilistic) defense in practice and had questions on the feasibility of this approach for large models. The authors argue that several modern and scalable methods exist for this, and mention they will clarify this further in their revised manuscript.

---

### Decision · Program_Chairs · 2025-01-22

Accept (Poster)